# A highly conserved core bacterial microbiota with nitrogen-fixation capacity inhabits the xylem sap in maize plants

Liyu Zhang[1,8], Meiling Zhang[1], Shuyu Huang[1], Lujun Li[2], Qiang Gao[3], Yin Wang[3], Shuiqing Zhang[4], Shaomin Huang[4], Liang Yuan[1], Yanchen Wen[1], Kailou Liu[5], Xichu Yu[5], Dongchu Li[1], Lu Zhang[1], Xinpeng Xu[1], Hailei Wei[6], Ping He [1], Wei Zhou[1], Laurent Philippot[7✉] & Chao Ai [1,8✉]

Microbiomes are important for crop performance. However, a deeper knowledge of crop-associated microbial communities is needed to harness beneficial host-microbe interactions. Here, by assessing the assembly and functions of maize microbiomes across soil types, climate zones, and genotypes, we found that the stem xylem selectively recruits highly conserved microbes dominated by Gammaproteobacteria. We showed that the proportion of bacterial taxa carrying the nitrogenase gene (*nifH*) was larger in stem xylem than in other organs such as root and leaf endosphere. Of the 25 core bacterial taxa identified in xylem sap, several isolated strains were confirmed to be active nitrogen-fixers or to assist with biological nitrogen fixation. On this basis, we established synthetic communities (SynComs) consisting of two core diazotrophs and two helpers. GFP-tagged strains and $^{15}$N isotopic dilution method demonstrated that these SynComs do thrive and contribute, through biological nitrogen fixation, 11.8% of the total N accumulated in maize stems. These core taxa in xylem sap represent an untapped resource that can be exploited to increase crop productivity.

[1] Key Laboratory of Plant Nutrition and Fertilizer, Ministry of Agriculture and Rural Affairs/Institute of Agricultural Resources and Regional Planning, Chinese Academy of Agricultural Sciences, Beijing 100081, PR China. [2] Hailun National Observation and Research Station of Agroecosystems, Key Laboratory of Mollisols Agroecology, Northeast Institute of Geography and Agroecology, Chinese Academy of Sciences, Harbin 150081, PR China. [3] Jilin Agricultural University, Changchun 130118, China. [4] Institute of Plant Nutrition, Resource and Environment, Henan Academy of Agricultural Sciences, 116 Garden Road, Zhengzhou 450002, China. [5] Jiangxi Institute of Red Soil, National Engineering and Technology Research Center for Red Soil Improvement, Nanchang 330046, China. [6] Key Laboratory of Microbial Resources Collection and Preservation, Ministry of Agriculture and Rural Affairs/Institute of Agricultural Resources and Regional Planning, Chinese Academy of Agricultural Sciences, Beijing 100081, PR China. [7] Université Bourgogne Franche-Comté, INRAE, AgroSup Dijon, Agroécologie, 21000 Dijon, France. [8] These authors contributed equally: Liyu Zhang, Chao Ai. ✉email: laurent.philippot@inrae.fr; aichao@caas.cn

Maize (*Zea Mays* L.) is grown worldwide and is of great importance for food production. In 2020, the global production of maize was estimated at 1.05 million thousand tons[1], ranking first in terms of yield production[2]. Several studies have shown that microbiomes have essential functions in maize performance. For example, field experiments in nitrogen (N)-depleted soil in Mexico indicated that the mucilage of aerial roots of a maize landrace was enriched with diazotrophs, and their fixation of atmospheric N contributed 29–82% of the N nutrition of maize[3]. Similar to the human microbiome[4], the plant microbiome is referred to as the second genome of the plant, and is crucial for plant health[5]. Thus, targeted manipulation of crop microbiomes represents a promising approach to maximise sustainable crop production in the future[6].

It is now widely accepted that plants and microorganisms can form complex co-associations governed by specific assembly rules[7]. As such, the recruitment and the selection of host-adapted microorganisms is of importance for crop health and nutrition[7,8]. It has even been suggested that plants and their associated microbiome are collectively forming holobionts and therefore should no longer be considered as standalone entities[9]. Recent studies have reported that a "core microbiota" exists within the plant, i.e., a subset of the plant microbiota that is reproducibly associated with a particular crop species across a wide range of scales[7,10]. These core microorganisms have a persistent relationship with the host, irrespective of the environmental conditions[11]. Since the core microbiota has traits for efficient colonisation, nutrient acquisition, and stress tolerance[12,13], it is likely to be particularly important for the biological functions of the host and may, therefore, affect crop health and nutrition. In recent years, the development and construction of synthetic communities (SynComs) has provided functional and mechanistic insights into microbe-host relationships and how these relationships influence plant fitness[7]. For example, inoculation of maize seedlings with a simplified and representative seven-strain SynCom resulted in a stronger biocontrol effect against the phytopathogenic fungus *Fusarium verticillioides* than inoculation with each strain separately, indicating a clear benefit to the host[14].

Previous studies have mainly characterised the core microbiota in the belowground compartments of plants[15,16], the phyllosphere[17], and seeds[18]. For example, Lundberg *et al.* initially identified core taxa in *Arabidopsis thaliana* roots as Actinobacteria and other specific families[15]. Grady *et al.* identified seasonally persistent and dynamic core phyllosphere microbiotas in perennial biofuel crops[17]. However, most studies on the assembly of the core microbiome and its collective functions in terms of host health and nutrition have overlooked the other plant compartments, such as vascular tissues. The vascular tissue acts as an effective long-distance transport system, which is driven by hydrostatic pressure gradients between the root and the shoot[19]. This driving force ensures smooth transport of solutes and signals among plant organs[20]. Moreover, the sizes of the holes in the perforated plates between xylem elements are large enough to allow passage of bacteria[21,22]. A few studies have reported that some systemic bacterial colonisers can spread to aboveground plant compartments through transpiration-driven xylem flow[22]. Anguita-Maeso *et al.* characterized the bacterial communities inhabiting the olive xylem sap by culture-dependent and independent approaches and found that *Sphingomonas* was the most representative genera[23]. Nevertheless, little is known about the functional relationship between the xylem-inhabiting microbiota and plant growth and development.

Here, we determined whether maize has core microbiomes in different plant compartments that can perform functions increasing host fitness. For this purpose, we investigated the microbiome of seven plant compartments in maize genotypes grown across a range of environmental conditions (i.e., different fertilisation regimes, soil types, climate zones). We found that the microbiota was more conserved in the xylem across soils than in the other plant compartments. We identified a core microbiota of 25 operational taxonomic units (OTUs) that was consistently detected in all xylem sap samples. Strains isolated from this core microbiota were capable of biological N-fixation (BNF). Their endophytic behaviour and contribution to maize N nutrition was verified by analyses of GFP-tagged strains and a $^{15}$N isotopic dilution method. Our study reveals details of an overlooked core microbiome inhabiting the xylem sap and how it contributes to the N nutrition of maize.

## Results

**Effects of soil type and fertilisation on the maize microbiome.** To detect differences in the maize microbiota among various plant compartments along the soil–maize continuum, we sampled plants from six long-term fertilisation experimental sites with three soil types (black soils, BSA; red soils, RSA; fluvo-aquic soils, FSA) across the middle temperate zone to the subtropical zone of China (Supplementary Data 1 and Supplementary Fig. 1). The soils in these areas have been fertilised for at least 29 years according to three regimes (no fertilisation, Control; chemical fertiliser N, phosphorus, and potassium, NPK; and organic manure plus chemical fertiliser, NPKM), which has led to drastic differences in soil fertility levels and bacterial communities (Supplementary Table 1 and Supplementary Fig. 2). Analyses of bacterial communities from the bulk soil (BS), rhizosphere soil (RS), root endosphere (RE), xylem sap (VE), stem endosphere (SE), leaf endosphere (LE), and phylloplane (P) by 16 S rRNA amplicon sequencing of the V5–V7 region suggested that plant compartments were the main driver of maize microbiome composition (plant compartments, $R^2 = 57.72\%$, $P < 0.001$; sites, $R^2 = 7.20\%$, $P < 0.001$; fertilisation treatments, $R^2 = 0.53\%$, $P < 0.001$, Fig. 1a). Moreover, an unconstrained principal coordinates analysis (PCoA) of data from the BS, RS, RE, VE, SE, LE, P samples showed that the beta-diversity of aboveground bacterial microbiomes differed from that of the root and soil microbiomes (Supplementary Fig. 2 and Supplementary Fig. 3). This trend was further captured by an independent permutational multivariate analysis of variance (PerMANOVA), which showed a gradually decreasing influence of site and fertilisation on bacterial communities from the soil to xylem sap (from 91.93% in BS to 28.42% in VE) (Fig. 1b; Supplementary Table 2). Alpha-diversity analyses also revealed significantly lower diversity of the microbiota in VE ($P_{FDR} < 0.05$) compared with that in BS and other plant compartments (Fig. 1c and Supplementary Fig. 4).

To further explore the influence of the environment on the plant microbiome, distance-decay relationships (DDRs) were analysed. We observed stronger geographic, edaphic, and climatic distance-decay of community similarity for RS and BS (Fig. 1d–f; Supplementary Table 3) than for the other compartments. However, such relationships were not observed in VE across geographic distance ($R^2 < 0.001$; $P = 0.601$) and climatic distance ($R^2 < 0.001$; $P = 0.237$). Non-significant correlations between the microbiome in VE and most soil physicochemical properties (e.g., total carbon, available phosphorus, soil water content) were confirmed by the Mantel test (Supplementary Fig. 5 and Supplementary Table 4). Together, these results indicate that, despite large edaphic and climatic differences across sites, the specific recruitment of bacterial taxa inhabiting VE was robust across genotypes and different environmental conditions.

**Changes in abundance and composition of maize-associated microbiomes along the soil-plant continuum.** We investigated

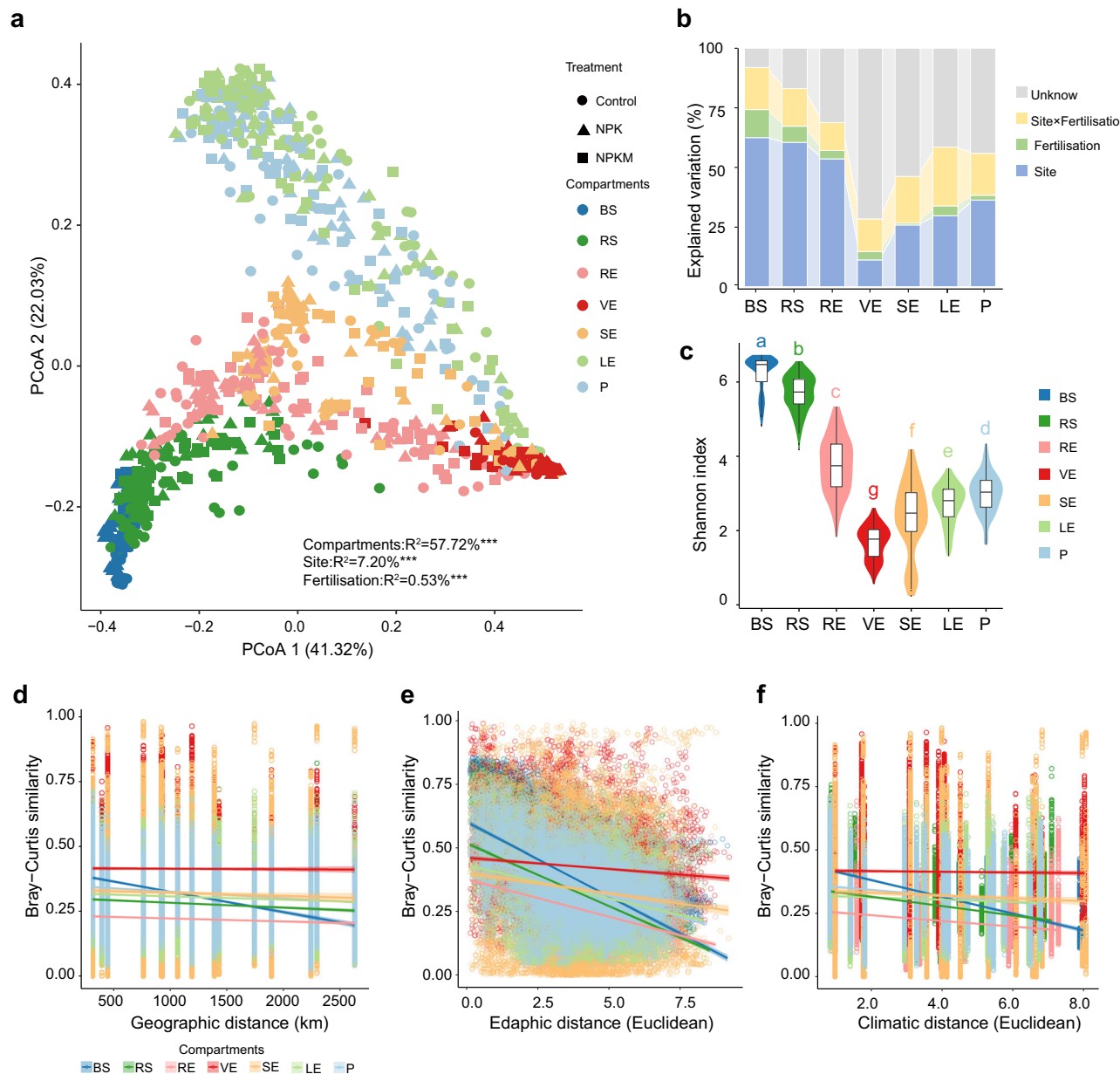

**Fig. 1 Effects of soil type and fertilisation on the maize microbiome. a** Unconstrained principal coordinates analysis (PCoA) with weighted unifrac distance across the whole dataset. (***$P$ < 0.001, PerMANOVA by Adonis). Control: no fertilisation; NPK: chemical fertiliser nitrogen, phosphorus, and potassium; NPKM: organic manure plus chemical fertiliser; BS: bulk soil; RS: rhizosphere soil; RE: root endosphere; VE: xylem sap; SE: stem endosphere; LE: leaf endosphere; P: phylloplane. **b** Effects of site, fertilisation treatments, and site × fertilisation on bacterial community structure in each compartment as tested by PerMANOVA. **c** Violin plot showing distribution of Shannon's index of the bacterial community in each compartment. Horizontal bars within boxes denote medians. Tops and bottoms of boxes represent 25th and 75th percentiles, and lines extend to the 1.5× interquartile range. Letters indicate statistical significance among groups using two-sided Wilcoxon test (adjusted $P$ < 0.05 by Benjamini and Hochberg method). The sample sizes are as follows: BS, 126; RS, 141; RE, 138; VE, 119; SE, 120; LE, 158; P, 152. **d–f**, Distance-decay curves showing Bray–Curtis similarity against geographic distances (**d**), edaphic distances (**e**), and climatic distances (**f**). Solid lines represent ordinary least-squares linear regressions. Source data and exact $P$ values are provided in the Source Data file.

the bacterial community composition along the soil–plant continuum, in particular, the steady-state composition of bacterial microbiomes in VE. We found that maize microbiomes in BS and RS consisted mostly of Acidobacteria, Actinobacteria, Gammaproteobacteria, Alphaproteobacteria, and Chloroflexi, accounting for 69.42%–85.89% of total relative abundance depending on soil types (Fig. 2a and Supplementary Data 3). Gammaproteobacteria were more abundant in RE than in BS and RS, and were predominant in stem microbiota (VE and SE). The average relative abundance of Gammaproteobacteria varied from 12.5% (BS),

29.2% (RS), 59.3% (RE), to 93.2% (VE), indicating that the transition in relative abundance profiles of host-associated lineages among plant compartments was more gradual than discrete.

Next, we used fast expectation-maximisation microbial source tracking (FEAST) to track the potential sources of plant bacterial microbiomes (Fig. 2b). The host showed a strong selection effect on the compositional pattern of microbial communities in habitats from the surface to the interior (Fig. 2b and Supplementary Fig. 6a). For example, 60.61% of the microbiome

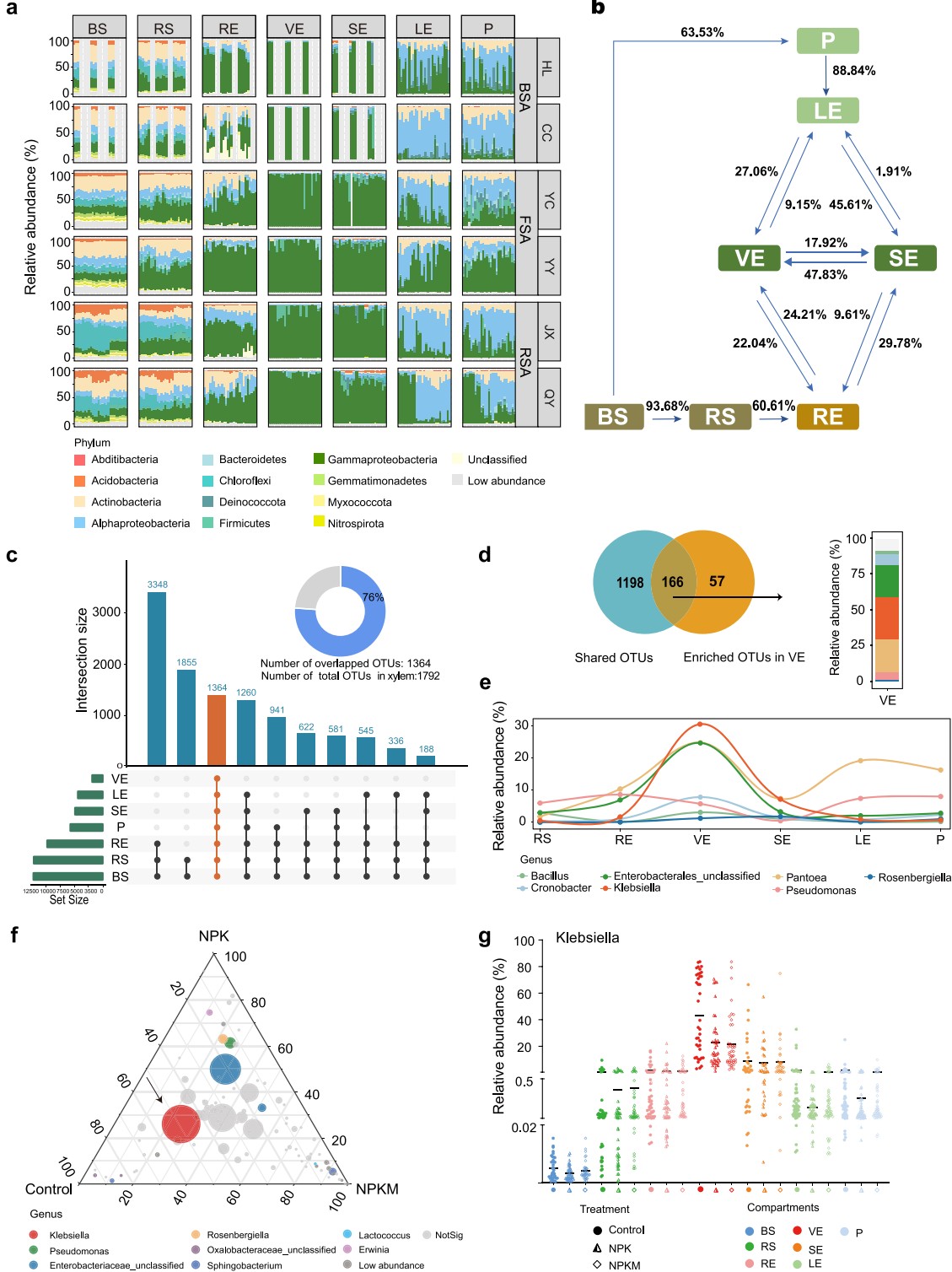

in RE was derived from RS. The VE acted as a linker between aboveground and belowground parts, with 24.21% of its bacterial microbiome originating from RE.

We further observed that the Gammaproteobacterial OTUs enriched in VE were dominated by Enterobacteriaceae, Erwiniaceae, and Pseudomonadaceae, while Burkholderiaceae was predominant in the SE population (Supplementary Fig. 6b, c). We detected 1364 OTUs that were shared among all plant compartments as well as the bulk soil, suggesting that they had the capacity to adapt to very different niches and could thrive

even in plant organs (Fig. 2c). Notably, this set of taxa accounted for 76% of the total OTU numbers in VE and 99% of the total relative abundance. Of these, 166 OTUs were enriched in VE compared with bulk soil, and they mainly belonged to the genera *Klebsiella* and *Pantoea* (Fig. 2d, e and Supplementary Data 4). A linear model analysis showed that the dominant *Klebsiella* in VE was more enriched in control soils (Fig. 2f), with a relative abundance up to 43.35%, than in fertilised soils (relative abundance of 22.95% in NPK and 21.83% in NPKM) (Fig. 2g). These results suggest that most taxa enriched in xylem likely

**Fig. 2 Maize bacterial community composition along the soil–plant continuum. a** Phylum-level distribution of bacterial communities across the whole dataset. Proteobacteria are shown at the class level. BS: bulk soil; RS: rhizosphere soil; RE: root endosphere; VE: xylem sap; SE: stem endosphere; LE: leaf endosphere; P: phylloplane. BSA: black soils; FSA: fluvo-aquic soils; RSA: red soils; HL: Hailun; CC: Changchun; YC: Yucheng; YY: Yuanyang; JX: Jinxian; QY: Qiyang. **b** Potential sources of maize bacterial communities as determined by FEAST. **c**, Vertical bars of upper plot show number of intersecting operational taxonomic units (OTUs) among plant compartments and soil, denoted by connected black circles below the histogram. Orange bars and circles represent OTUs that overlap among seven compartments, horizontal bars show OTUs set size. Inset donut plot displays overlapping OTUs (accounting for 76% of total OTUs in xylem sap). **d** Venn diagram showing overlap between shared and enriched OTUs, and their relative abundance at genus level. **e** Abundance profile of OTUs overlapping along plant compartments as denoted by curves. **f** Ternary plots of OTUs in xylem sap across three fertilisation treatments. Size of each point represents relative abundance of OTU. Position is determined by the contribution of three fertilisation treatments to the total relative abundance, proximity to that vertex indicates enrichment of that OTU in that fertilisation treatment. Colours of circles correspond to different genera. Grey circles indicate OTUs with no significant differences in abundance. **g** Distribution of *Klebsiella* in each compartment under fertilisation treatments. Horizontal black bars represent the mean of each group. Source data are provided in the Source Data file.

originated from the soil, gradually moved into different plant compartments, and eventually thrived in the stem xylem, where they may play a role in maize nutrition.

**Xylem-inhabiting bacterial communities enriched functional roles related to N cycling.** Given the key role of the stem xylem as the channel for materials transport between the aboveground and belowground compartments, we performed a functional annotation of prokaryotic taxa (FAPROTAX) analysis to generate putative functional profiles of the microbiotas inhabiting xylem and other compartments based on their community composition (Fig. 3a). The microbiotas in the aboveground and belowground compartments were separated into two clear clusters in terms of their putative functional profiles. Most functions were more abundant in root-associated environments than in aboveground compartments, for example, cellulolysis, predatory/exoparasitic, chitinolysis, and ureolysis. However, potential functions related to fermentation and N-cycling were overrepresented in VE. Specifically, the number of OTUs involved in N-cycling was approximately 2-times higher in VE than in LE, and 1.5-times higher than in SE (Fig. 3b). The number of these OTUs in VE was also significantly ($P_{FDR} < 0.05$) higher in control soils than in the two long-term fertilised soils (NPK and NPKM), suggesting that these plant-recruited bacteria were more likely to play a critical role in N availability in plants under soil nutrient stress (Fig. 3b). However, like other tools assigning ecological functions based on 16 S data, FAPROTAX has some limitations related to the number of reference strains and the affiliation of functions that are poorly conserved; therefore, our FAPROTAX prediction needs to be confirmed by other methods.

To test this hypothesis, we performed triple-qPCR (see Methods) to determine the ratio of the nitrogenase gene (*nifH*) to the 16 S rRNA gene, as a proxy of the relative N-fixation potential of the bacterial community. As expected, this ratio was significantly higher ($P_{FDR} < 0.05$) in VE than in the other three aboveground compartments (SE, LE, and P) in control soil only (Fig. 3c).

**Identification and isolation of xylem-inhabiting core bacterial communities.** Abundance–occupancy analyses identified 25 OTUs (occupancy of 1) that were persistent in xylem in all samples (Fig. 4a and Supplementary Data 5). These taxa were dominated by Gammaproteobacteria (89.42%), Firmicutes (4.10%), Alphaproteobacteria (0.18%), and Actinobacteria (0.13%), with an overrepresented OTU belonging to the *Klebsiella* genus (29.2%) (Supplementary Fig. 7). Their ubiquitous presence and high abundance indicated that these taxa were core members of maize-associated bacterial communities. Next, we isolated 969 pure bacterial isolates from the xylem sap using three media, representing a total of 109 unique strains belonging to four phyla

and 23 families (Supplementary Data 6). Further characterisation of these isolates revealed that nine out of 109 strains grew on N-free medium, harboured the *nifH* gene, and showed high nitrogenase activity (Fig. 4b). We then cross-referenced the V5–V7 region of the 16 S rRNA gene of the 109 isolated strains against the highly abundant OTUs (> 0.01%) in VE. We found that these isolated strains shared sequence identity with 14 of the 25 core OTUs in VE, which accounted for 81.33% of the total bacterial abundance in VE (Fig. 4c, Supplementary Fig. 7 and Supplementary Data 7). Further, two of the 14 strains displayed remarkably high nitrogenase activity (*Klebsiella variicola* MNAZ1050 and *Citrobacter* **sp**. MNAZ1397, their combination hereafter referred to KC), and harboured the entire *nif* gene cluster in their genomes (Fig. 4d).

To explore whether the non-N-fixers in the core VE microbiota can assist in biological N-fixation, we mixed 12 non-N-fixers with diazotrophs and determined the total nitrogenase activity of each combination. We found that, albeit to different extents, the nitrogenase activity of each combination was higher than that of the diazotrophs alone, especially for the combinations of KC + *Acinetobacter* **sp**. ACZLY512 and KC + *Rosenbergiella epipactidis* YCCK550 (Fig. 4e). By sequencing the genomes of these non-N-fixing core strains, we found that the total reads of genes encoding major enzymes involved in the tricarboxylic acid (TCA) cycle was higher in *Acinetobacter* **sp**. ACZLY512 and *R. epipactidis* YCCK550 than in other strains (Supplementary Fig. 8a and Supplementary Data 8). Therefore, these non-N-fixers may have strong respiratory metabolism, thereby creating a micro-oxygen environment conducive to biological N-fixation. This was further confirmed by acetylene reduction assays (ARA) under initial atmospheric oxygen levels (see Methods), which showed that the nitrogenase activity of the diazotrophs was no longer inhibited in the presence of air after adding *Acinetobacter* **sp**. ACZLY512 (Supplementary Fig. 8b). We also found that there were more functional gene reads involved in ammonia assimilation in the two non-N-fixers (Supplementary Fig. 8c and Supplementary Data 8). This implied that they might facilitate N-fixation, because the activation and transcription of nitrogenase genes is positively regulated by ammonia comsumption[24,25].

**Successful colonisation and N-fixation by synthetic communities in xylem vessels.** We further verified the endophytic behaviour and N-fixation function of the core strains by constructing and analysing synthetic communities (SynComs). The SynComs consisted of two N-fixing bacteria (*K. variicola* MNAZ1050 and *Citrobacter* **sp**. MNAZ1397) and two non-N-fixing strains that may help to fix N (*Acinetobacter* **sp**. ACZLY512 and *R. epipactidis* YCCK550), which together accounted for 34.61% of the total OTU abundance in the VE bacterial microbiota (Fig. 4e). At 63 days after inoculation onto maize plants, the successful colonisation of soil-derived SynComs in the maize

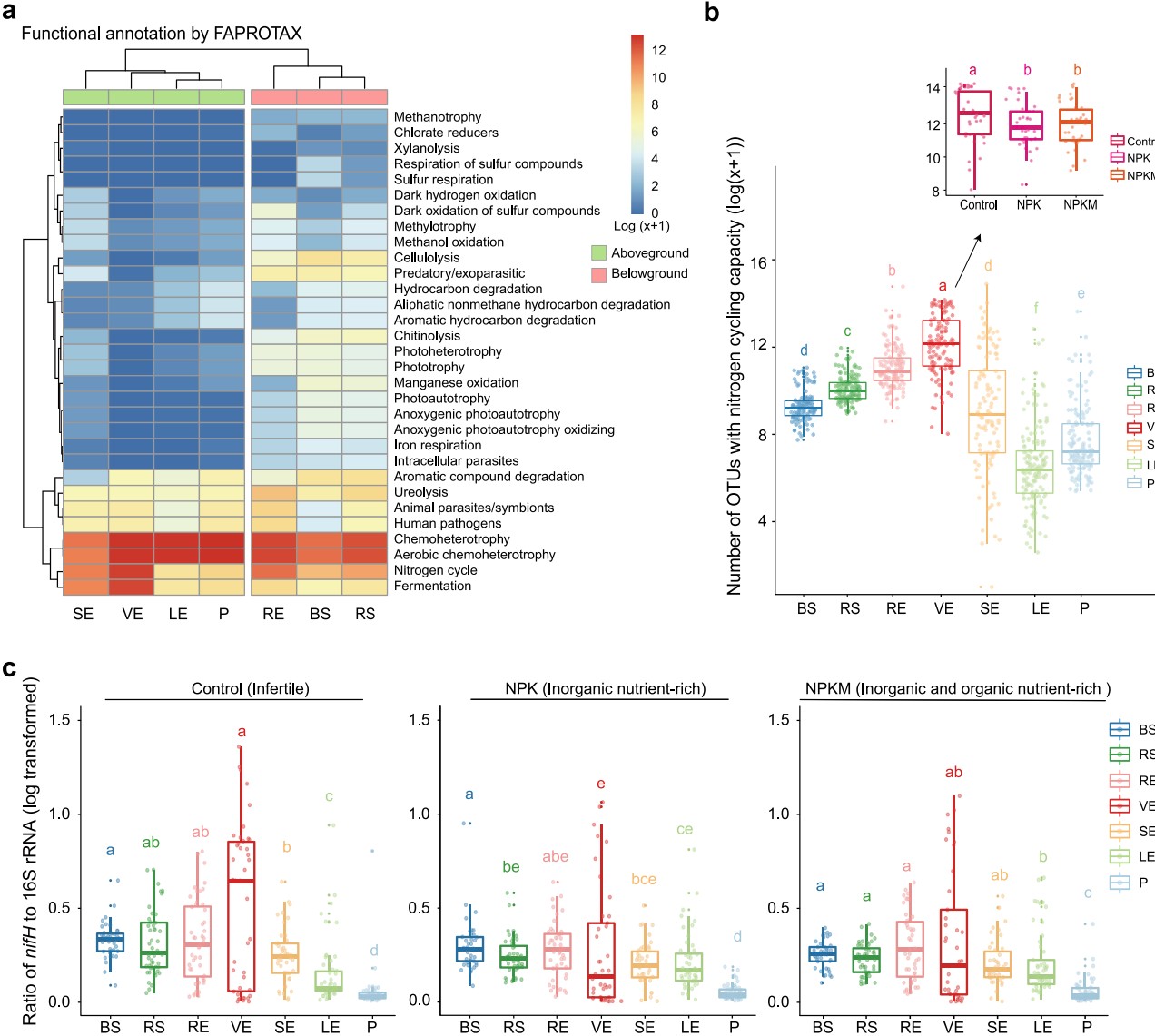

**Fig. 3 Functional characteristics of xylem-inhabiting bacterial communities. a** Heatmap of metabolic and ecological functions of bacteria based on FAPROTAX prediction. Data are based on OTUs occurrence (number of OTUs capable of each function) in soil and plant compartments (log(x + 1) transformed). **b** Number of OTUs capable of nitrogen cycling in soil and plant compartments based on FAPROTAX prediction. Boxplot on the upper right shows the number of OTUs capable of nitrogen cycling in fertilisation treatments. The sample sizes are as follows: BS, 126; RS, 141; RE, 138; VE, 119; SE, 120; LE, 158; P, 152; Control, 38; NPK, 38; NPKM, 42. **c** Ratio of *nifH* gene to 16 S rRNA in each compartment under fertilisation treatments, reflecting the relative N-fixation potential of bacterial communities. The sample sizes are as follows, Control: BS, 41; RS, 48; RE, 48; VE, 39; SE, 42; LE, 52; P, 50; NPK: BS, 42; RS, 48; RE, 48; VE, 38; SE, 41; LE, 48; P, 52; NPKM: BS, 41; RS, 48; RE, 47; VE, 41; SE, 41; LE, 53; P, 54. Boxplot (**b** and **c**) displays the median and interquartile range. Upper and lower whiskers extend to data no more than 1.5× the interquartile range from the upper and lower edge of the box, respectively. Letters indicate statistical significance among groups using two-sided Wilcoxon test (adjusted $P < 0.05$ by Benjamini and Hochberg method). Source data and exact $P$ values are provided in the Source Data file.

aboveground compartments was detected. As shown in the confocal laser scanning microscope (CLSM) images, only the inoculated maize plant had bright green fluorescent signals from the GFP-tagged SynComs within the stem and xylem sap (Fig. 5a and Supplementary Fig. 9).

The contribution of biological N-fixation of the inoculated SynComs to maize N-nutrition was measured by the $^{15}N$ isotopic dilution method. The maize plants inoculated with the SynComs had a higher proportion of N from the air (lower $\delta^{15}N$ values) than did the uninoculated plants. The contribution of BNF to total accumulated N in maize roots, stems, and leaves was 4.01%, 11.81%, and 0.46%, respectively (Table 1). Characterisation of

maize plant phenotypes revealed that, compared with uninoculated control plants, those inoculated with the SynComs showed significantly higher root dry weight (Fig. 5b, c and Supplementary Fig. 9) and root-to-shoot ratio (Fig. 5d, independent samples T-test, $P < 0.01$).

## Discussion

The conductive property of plant xylem vessels makes it a highly dynamic endosphere[21]. Xylem vessels with open perforation plates function as a free-flowing pipe, thereby allowing endophytes to spread longitudinally from the rhizosphere to other plant tissues[22,26]. However, the importance of the xylem in the

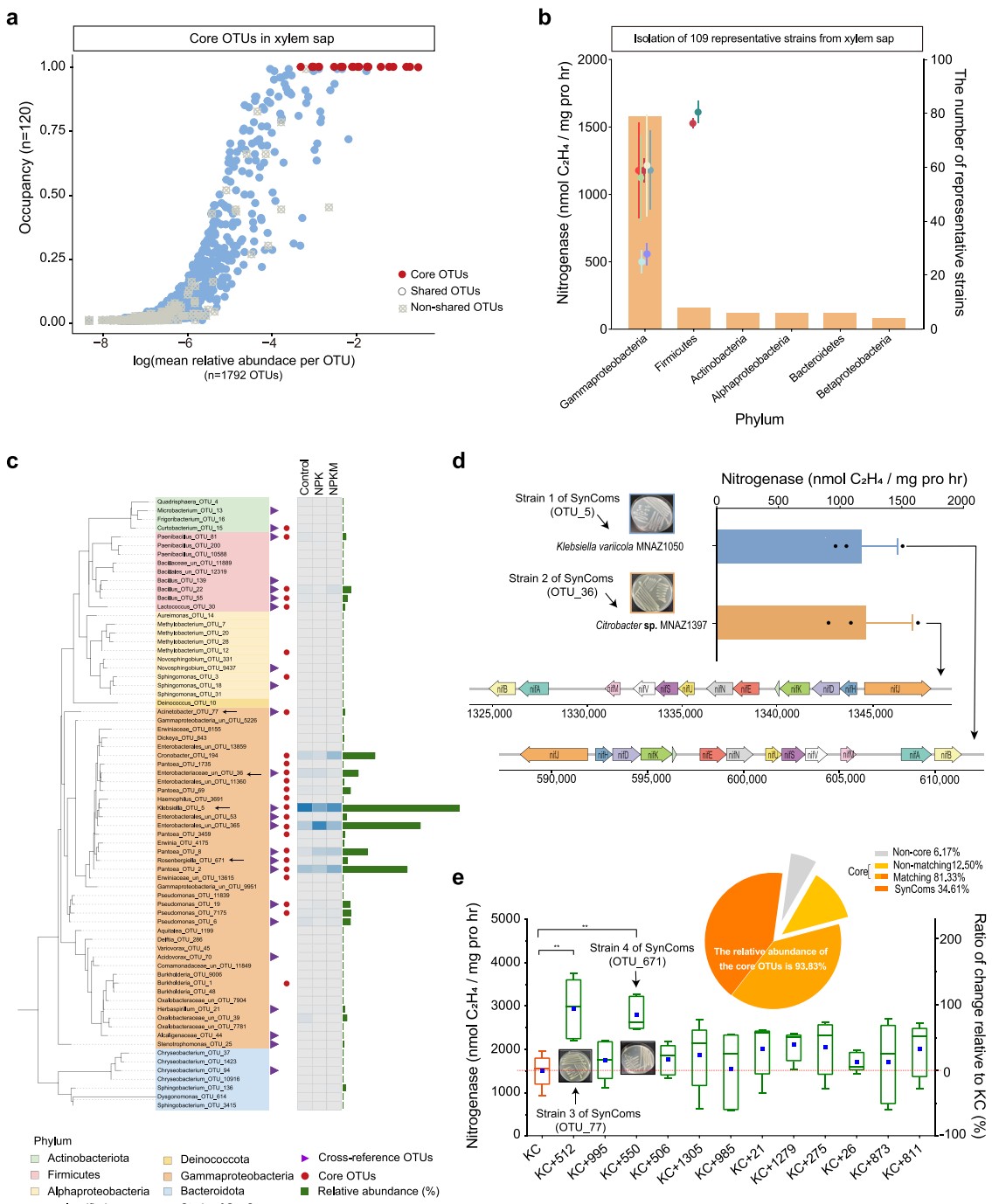

**Fig. 4 Identification and isolation of xylem-inhabiting core bacterial communities. a** Occupancy–abundance curves for xylem communities. X-axis displays the log-transformed mean relative abundance of each OTU, y axis displays percentage of samples in which each OTU was detected. OTUs with 1 occupancy are shown in red (core taxa); blue circles represent taxa shared among soil and plant compartments; grey squares indicate non-shared OTUs. **b** Bar graph depicting number of representative strains at phylum or class level. Points indicate mean nitrogenase activity of diazotrophs at each phylum or class level; error bars indicate standard deviation ($n = 5$ for each strain). **c** Cultivation-dependent coverage of OTUs in xylem. Inner column represents OTUs in xylem with relative abundance >0.01%. Red circles represent core OTUs, which are the same as Fig. 4a. Purple triangles mark OTUs that matched with culturable bacteria. White and blue heatmap represents differences in relative abundance among fertilisation treatments. Green bar graph represents relative abundance of OTUs with a maximum value of 29.2%. **d** Nitrogenase activity of two core strains (MNAZ1050 and MNAZ1397). The bars indicate mean values and the error bars indicate standard deviation ($n = 3$). Arrows mark *nif* genes with locations shown below arrows. **e** Nitrogenase activity of 12 combinations relative to KC (KC denotes combination of two diazotrophs, MNAZ1050 and MNAZ1397). Blue rectangles represent the ratio of change relative to KC. Horizontal bars within boxes denote medians. Tops and bottoms of boxes represent 25th and 75th percentiles, and lines extend to 1.5× interquartile range. The statistical analyses were performed using a two-sided T-test ($n = 5$). *P* values are indicated by *, i.e., ** represents *P* < 0.01. Pie plot shows total relative abundance of each group (core OTUs, noncore OTUs, matching OTUs, non-matching OTUs, SynComs (synthetic communities)). Source data and exact *P* values are provided in the Source Data file.

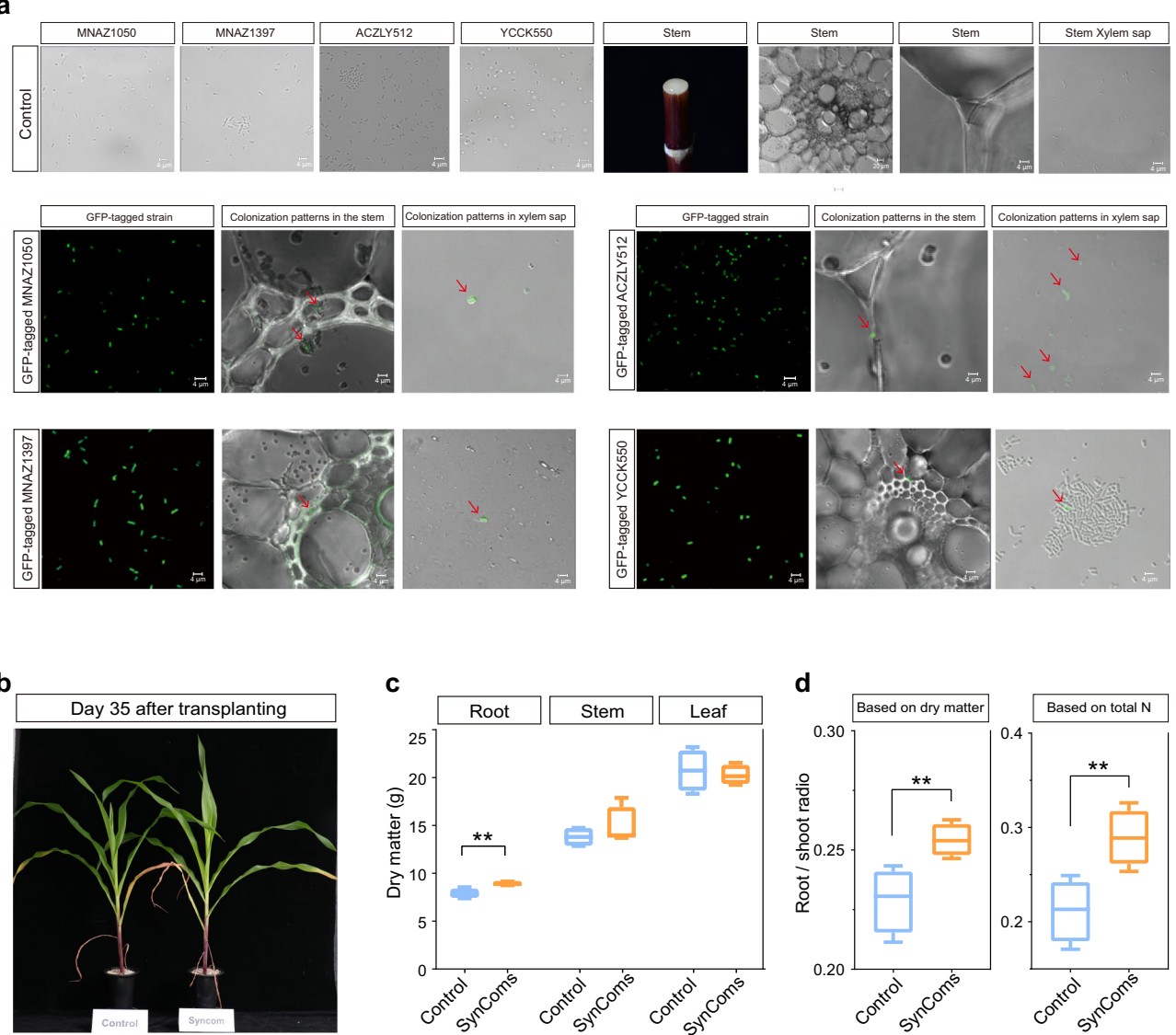

**Fig. 5 Colonisation and nitrogen fixation by SynComs in xylem of maize plants. a** Potted plant experiment with GFP-tagged SynComs. SynComs with no GFP tags served as control. Images are confocal laser scanning microscopy (CLSM) micrographs. Four biological replicates showed the similar results. **b, c, d** Potted plant experiment with growth mixture containing ¹⁵N. Growth profile (**b**) at 35 days after transplanting after inoculation with control and SynComs, dry matter of root, stem and leaf (**c**), and root/shoot ratio (**d**) of maize plants at 65 days after transplanting after inoculation with control and SynComs. Horizontal bars within boxplot represent medians. Asterisk indicates a significant difference at $P < 0.01$. Boxplot displays the median and interquartile range. Upper and lower whiskers extend to data no more than 1.5× the interquartile range from upper and lower edge of the box, respectively. The statistical analyses were performed using a two-sided T-test ($n = 4$). $P$ values are indicated by *, i.e., ** represents $p < 0.01$. Source data are provided in the Source Data file.

plant microbiome assembly has been overlooked. In the present study, while the root- and leaf-associated bacterial communities varied significantly with environmental factors as previously reported[27,28], the stem xylem recruited highly conserved taxa independent of geographic ($P = 0.601$) and climatic ($P = 0.237$) distances. A FEAST analysis showed that 60.61% of bacteria in the root endosphere were derived from rhizosphere soils; 24.21% of bacteria in the stem xylem were derived from the root endosphere and 9.15% of bacteria in the leaf endosphere were derived from xylem-inhabiting endophytes (Fig. 2b). It can therefore be assumed that these facultative endophytes positively selected by the host have the physiological plasticity to thrive in different plant compartments, with functionally important roles[29,30]. Accordingly, the proportion of bacterial taxa harbouring N-fixation genes was larger in the xylem than in other plant compartments. For example, the ratio of *nifH* to 16 S rRNA in xylem was about twice that in the root endosphere and four times that in the leaf endosphere, especially in control soils where this function is likely to be more important because plants are under nutrient stress. This is also confirmed by the fact that the average relative abundances of two core N-fixers in SynCom (*K. variicola* MNAZ1050 and *Citrobacter* **sp**. MNAZ1397) were significantly higher in the control soil than in two fertilised soils (NPK and NPKM) (Supplementary Fig. 10). Therefore, our results confirm those of previous studies, and suggest that the beneficial effects of endophytic bacteria could be as important as those of the microorganisms present in the rhizosphere[29,31].

We further conducted an abundance–occupancy pattern analysis to identify the core subset of bacterial communities in xylem. This approach provides ecological insights into which taxa are

**Table 1 Contribution of biological N-fixation by SynComs to N nutrition of roots, stems, and leaves of maize plants grown in medium containing $^{15}$N.**

| Treatment | | δ$^{15}$N value | BNF (% Ndfa) | N$_2$-fixed (mg) |
|---|---|---|---|---|
| Root | Control | 104,897.96 | – | – |
|  | SynComs | 100,693.23 | 4.01 | 2.55 |
| Stem | Control | 106,646.60 | – | – |
|  | SynComs | 94,046.58 | 11.81 | 9.72 |
| Leaf | Control | 103,159.93 | – | – |
|  | SynComs | 102,689.50 | 0.46 | 0.63 |

Values represent mean of four replicates, δ$^{15}$N value: percent atom excess $^{15}$N, % Ndfa: percent N derived from atmosphere, N-fixed: total N derived from atmosphere.

most important for the plant[32]. We used a conservative threshold for the occupancy analysis, so that members entering the xylem due to chance events such as wounds and cracks would be excluded. Both the occupancy and abundance patterns indicated the presence of a core microbiome consisting of 25 OTUs with high abundance, consistent with previous studies that identified core microorganisms across a wide of sites[10,33]. The fact that the samples were collected from very diverse sources (six different long-term fertilisation experiments, from sites spanning three climate zones, and encompassing multiple maize genotypes) provides further evidence for the persistent association between the core subset and its host. The isolation of endophytic diazotrophic bacteria from root, stem, and leaf tissues of maize cultivars has been demonstrated in a previous study[34]. Here, concurrent culture-dependent methods yielded isolates that were cross-referenced against the core OTUs. Among these isolates, one strain mapping to the most abundant core OTU (29.2%) was identified as *K. variicola* and contained the entire *nif* gene cluster with nitrogenase activity of up to 1180 nmol C$_2$H$_4$ (mg protein hr)$^{-1}$. A previous study reported that *Klebsiella* is present as an endophyte in maize stem and root tissue, and quantified its dinitrogenase reductase activity inside maize tissue after adding an exogenous carbon source[35]. Our results indicate that *K. variicola* is dominant within the maize core microbiota across different soil types, climate conditions, and maize genotypes, suggesting that its key beneficial role in maize might be more general. Furthermore, we found that the microbial communities in xylem include core non-N-fixers such as *Acinetobacter* **sp**. ACZLY512 and *R. epipactidis* YCCK550 with the potential to assist in N-fixation. The genome analysis of these strains provided a possible explanation for this mutually beneficial symbiosis, with both genomes showing a higher abundance of gene clusters potentially involved in the TCA cycle and ammonia assimilation. Bacteria with the ability to perform BNF are known to be sensitive to the concentrations of fixed N and external oxygen[24]. Thus, we speculate that these core non-N-fixers might assist in N-fixation by regulating N and oxygen concentrations in the xylem microenvironment where the diazotrophs are located.

Finally, we established SynComs consisting of two diazotrophs and two helper strains. Our results confirmed the endophytic lifestyle and N-fixation capacity of the SynComs *in planta*. The contribution of BNF to total N accumulation was higher in maize stems (11.81%) than in roots (4.01%) and leaves (0.46%), demonstrating that SynComs do thrive and contribute to plant N-nutrition in the stem xylem after being recruited from the soil. Interestingly, the SynComs also increased root dry biomass (Fig. 5b, c), suggesting that there is a multi-mechanism synergistic interaction between the core endophytes and their host. This is supported by the fact that the genomes of *K. variicola* MNAZ1050 and *Acinetobacter* **sp**. ACZLY512 contain genes

involved in IAA biosynthesis and ethylene biosynthesis, respectively. Thus, hormones produced by these strains may promote root growth. Such synergistic effects of the bacterial consortium were previously observed in other plant-bacterial interactions. For example, Santhanam *et al*. showed that a five-strain SynCom was able to effectively protect *Nicotiana attenuata* against fungal sudden-wilt disease via complementary traits, while the individual strains or smaller combinations of them did not[36]. Overall, our results highlight the limitation of incorporating only the desirable traits into the design of SynComs since this method might overlook helper strains that could improve the overall beneficial activity of the SynComs.

In summary, our data indicate that maize specifically recruit a core microbiota in the xylem sap that is conserved across environmental conditions as well as genotypes. This core microbiota contributes to plant N-nutrition through BNF and promotes root development. Further research is needed to explore the mechanisms and functions involved in the association between the xylem sap microbiota and its host. This core microbiota could represent a promising resource to develop alternative microbial biotechnologies to enhance crop performance in sustainable agriculture.

## Methods

**Site description**. The six long-term fertilisation field experiments were located across a latitudinal gradient in China from north to south, spanning three climate zones from the middle temperate zone to the subtropical zone. These sites were chosen to represent the three main agricultural production areas in China. The soils at the sites were black soils (Udolls or Typic Hapludoll according to USDA Soil Taxonomy, BSA) in Hailun (Heilongjiang Province) and Changchun (Jilin Province); fluvo-aquic soils (Aquic Inceptisol according to USDA Soil Taxonomy, FSA) in Yucheng (Shandong Province) and Yuanyang (Henan Province); and red soils (Ultisols according to USDA Soil Taxonomy, RSA) in Jinxian (Jiangxi Province) and Qiyang (Hunan Province). The two most distant sites (from Hailun to Qiyang) were more than 2,500 km apart, and the two closest test sites (from Yucheng to Yuanyang) were at least 300 km apart. Three treatments, i.e., no fertiliser (Control), chemical fertiliser N, P, and K (NPK), and organic manure plus chemical fertiliser (NPKM), have been applied in triplicate plots in each field for 29 years or more. Climate data corresponding to the sampling site coordinates were obtained from the China Meteorological Data Network (http://data.cma.cn/). Further details of experimental sites are provided in Supplementary Data 1.

**Sample collection**. Sampling was performed during the silking-maturity period of maize in 2019 and 2020, with the exact date of sampling depending on the developmental stage of plants at each location (Supplementary Data 2). In the FSA and RSA soils, three individual maize plants were selected from each subplot (a total of 27 maize plants per site). From each maize plant, we collected the following compartments in the field: mixed leaves, xylem sap, stem, roots, bulk soil. The mixed leaves sample consisted of the 2nd, 4th, and 6th leaves, which were removed from the plant stem using ethanol-sterilised scissors. To collect xylem sap, a proxy for xylem, we cut off the stem mid-way between the 2nd and 3rd node from the base of the plant, and sterilised absorbent cotton in sterilised bags was placed on the cut end of the shoot (see Supplementary Movie 1 for details of this procedure). Meanwhile, we inserted a steel stick (sterilised, 2 cm diameter, 30 cm long) into the soil at each subplot to simulate the collection of xylem sap and check for contaminants during the field operations (Supplementary Fig. 11). The stem sample consisted of the upper region between the 2nd and 3rd nodes, collected into sterilised bags. To collect root samples, we shook whole roots vigorously to remove all loose soil. The roots and root-adhered soil particles were collected for further separation of the roots and rhizosphere soil in the laboratory. The bulk soil sample was collected from between the rows of maize plants. At the sites with BSA soils, we only collected one individual maize plant from each subplot (a total of nine maize plants per site), because these two long-term experiments have strict requirements for sampling to avoid large-scale damage to the entire test field. Thus, for each plant, the 1st and 2nd, the 3rd and 4th, and the 5th and 6th leaves were collected as three replicates. Similarly, fine roots (< 2 mm) and thick roots (> 2 mm) were collected separately. Except for this difference, the other operations were the same as those at the other four experimental sites. All samples were placed on ice for transport and further processing within 48 h. The soil parameters of pH, total C (TC) and N (TN), ammonium (NH$_4$$^+$) and nitrate (NO$_3$$^-$), and soil available P (AP) and K (AK) are listed in Supplementary Table 1.

**Sample processing**. To recover the xylem sap absorbed in the cotton, each cotton ball was placed into a 50-mL sterile centrifuge tube with a filter and centrifuged at

6000 × *g* for 5 min. The collected sap was divided into two parts; one part was used for bacterial isolation, and the other part (stored at −80 °C) was used for culture-independent bacterial 16 S rRNA gene profiling.

**Processing of root-associated samples**. We used a modified protocol[15] to separate the microbiome living on the plant surface (epiphytes) from the microbiome living within the plant (endophytes). Briefly, 5 g root tissue was weighed into a 100 mL conical flask containing 80 mL sterile PBS and 5 μL Tween 80. The mixture was vortexed, and the liquid was collected as the rhizosphere (root epiphyte) sample. To extract rhizosphere DNA, the sample was centrifuged at 10,000 × *g* for 5 min, and then 500 mg of the resulting tight pellet containing fine sediment and microorganisms was placed in a Lysing Matrix E tube (supplied in the FastDNA™ Spin Kit for Soil). To obtain endophytes from the root samples, the roots were washed with fresh PBS until the buffer was clear after vortexing. The roots were then sonicated using an ultrasonic cell disruptor (Scientz JY 88- IIN, Ningbo Scientz Biotechnology Co., Ltd., Zhejiang, China) at a low frequency for 10 min (30-s bursts followed by 30-s rests).

**Processing of leaf-associated samples**. The method for washing leaf samples was similar to that used to wash the roots, except for an additional step before collecting the phyllosphere. Each leaf sample (5 g) was added to a conical flask containing sterile PBS, which was subjected to two 5-min treatments in an ultra-sound bath (25 °C, 40 KH$_Z$), with vortexing for 30 s between the two ultra-sonication treatments. This procedure released most of the phyllosphere microbes from the leaves. Then, the filtrate was collected and the leaves were washed again using the above procedure. Phyllosphere samples were collected by centrifugation (at 10,000 × *g* for 20 min) of the accumulated filtrate and were resuspended in 1 mL sodium phosphate buffer (FastDNA™ Spin Kit for Soil) before being transferred to Lysing Matrix E tubes (FastDNA™ Spin Kit for Soil). Finally, the leaf and stem samples were washed and sonicated in the same way as the roots. Sonicated root, leaf, and stem samples were snap-frozen in liquid N$_2$ and stored at −80 °C until analysis.

**DNA extraction, PCR amplification and sequencing**. Total DNAs were extracted from the aforementioned samples with a FastDNA™ Spin Kit for Soil (MP Biomedicals, Solon, OH, USA) following the manufacturer's instructions. The DNA concentration and purity were measured using a NanoDrop2000 spectrophotometer (NanoDrop2000, Thermo Fisher Scientific, Waltham, MA, USA). The DNAs extracted from soils (bulk and rhizosphere soil) were diluted 10-fold. For 16 S rRNA gene libraries, the V5–V7 region was amplified using the primers 799 F and 1193 R (Supplementary Table 5). Each DNA template was amplified in triplicate (together with a water control) in a 25-μL reaction volume. The PCR conditions were as follows: 12.5 μL 2× EasyTaq PCR SuperMix (TransGen Biotech, Beijing, China), 1.25 μL forward primers (10 μM), 1.25 μL barcoded reverse primers (10 μM), 1.25 μL template DNA, and 8.75 μL ddH$_2$O. The PCR amplification program was as follows: 94 °C for 3 min; 28 cycles of 94 °C for 30 s, 55 °C for 30 s, 72 °C for 90 s; and 72 °C for 90 s. The products were stored at 4 °C until use. After mixing the triplicate PCR products of each sample, the bacterial 16 S rRNA gene amplicons were extracted from a 1% agarose gel using a Gel Extraction Kit (Omega Bio-tek Inc., Norcross, GA, USA). The DNAs were measured using a Quant-iT™ PicoGreen™ dsDNA Assay kit (Thermo Fisher Scientific) and pooled in equimolar concentrations. Sequencing libraries were generated using an Illumina TruSeq DNA PCR-Free Library Preparation Kit (Illumina, San Diego, CA, USA) and were sequenced on the NovaSeq-PE 250 platform (Illumina).

**16 S rRNA gene amplicon sequence processing**. Paired-end reads were checked by FastQC v.0.10.1[37] and merged using the USEARCH 11.0.667[38] fastq_mergepairs script. Reads were assigned and demultiplexed to each sample according to the unique barcodes by QIIME 1.9.1[39]. After removing barcodes and primers, low-quality reads were filtered and non-redundant reads were identified using VSEARCH 2.12.0[40]. Unique reads with ≥ 97% similarity were assigned to the same OTU. Representative sequences were selected using UPARSE[41] and the classify.-seqs command in mothur[42] was used to taxonomically classify each OTU with reference to the SILVA 138 database[43]. The OTUs classified as host plastids, cyanobacteria, and others not present in our samples were removed from the dataset.

**Statistical analysis**. Alpha and beta-diversity analyses were conducted with R script as described in previous studies[44] and on the QIIME2[39] platform. PerMANOVA analyses were performed using the 'Adonis' function implemented in the vegan package[45] of R. We used the microbial source-tracking method FEAST[46] to determine the potential origin of the microbiota inhabiting the various compartments of maize plants. Bacterial OTUs were assigned into multiple functional groups using FAPROTAX v.1.2.1[47]. All analyses were conducted in the R Environment[48] except for beta-diversity analyses. All plots were generated with ggplot2[49] and GraphPad Prism 8.0.0 (GraphPad Software, San Diego, CA, USA, www.graphpad.com).

**Distance-decay relationship analyses**. We conducted distance-decay relationship analyses to assess the relationship between the similarity of communities in individual plant compartments and spatial distance, edaphic distance, and climatic distance. We used the Geosphere package to calculate the geographic distances in km from the latitude and longitude coordinates, and calculated edaphic and climatic distances separately as the Euclidian distance. We used the 'vegdist' function in the vegan package to calculate the Bray–Curtis similarity of microbial communities. Here, the variation in the slope of distance decay reflects the degree to which the similarity of microbial communities in plant compartments varies with environmental distances. The relationships between the Bray–Curtis similarity of each compartment and specific soil or climatic factors were determined by calculating Pearson's correlation (r) values. The significance of r values was assessed with the Mantel test implemented in the vegan package.

**Differential abundance testing**. A negative binomial generalised linear model was implemented with the edgeR package[50] to detect differences in OTU abundance among samples. We compared individual plant compartments (RS, RE, VE, SE, LE, and P) against bulk soil (BS) and conducted pairwise comparisons among fertilisation treatments. For each comparison, after constructing the DGEList object and filtering out low counts, the calcNormFactors function was used to obtain normalisation factors and the estimateDisp function was used to estimate tagwise, common, and trended dispersions. We then used the glmFit function to test the differential OTU abundance. The corresponding *P* values were corrected for multiple tests using FDR with α = 0.05.

**Core taxa selection**. We used the UpSetR package[51] to visualise the OTUs that overlapped among all plant compartments and soils. The overlapping OTUs were defined as those detected in at least one sample from each compartment. We further identified the union of overlapped OTUs and enriched OTUs in the xylem using the EVenn online tool[52]. Importantly, abundance–occupancy analyses were conducted to identify the core OTUs across environmental gradients. We calculated occupancy with the most conservative approach, which restricted the core to only those OTUs that were detected in all xylem sap samples (i.e., occupancy=1).

**Triple-qPCR to verify FAPROTAX results**. To detect contamination with plant organelles, sample DNAs were amplified using the universal primer pair 799 F/1193R[53], which amplifies both bacterial 16 S and mitochondrial 18 S rRNA but not chloroplast sequences; the mitochondrial-specific primer pair mito1345F/mito4130R[54,55], which only amplifies mitochondrial 18 S rRNA, and the *nifH* gene primers PolF/PolR[56] (Supplementary Table 5). To reflect the potential N-fixation of bacterial communities in each plant compartment, the following ratio was calculated:

$$\text{Relative nitrogen fixation potential} = \frac{nifH \text{ gene}}{\text{bacterial 16S and mitochondrial 18S rRNA} - \text{mitochondrial 18S rRNA}} \tag{1}$$

All qPCR assays were run on a QuantStudio 6 Flex using SYBR Green Pro Taq HS Premix (Accurate Biotechnology, Changsha, China) in a 20 μL volume containing 200 nM of each primer and approximately 50 ng DNA per reaction. All three primer sets were amplified in a three-step qPCR[57] run at 95 °C for 15 s, 55 °C for 30 s and 72 °C for 40 s for 40 cycles followed by a melting curve analysis. The amplification efficiency varied from 83 to 117%.

**Isolation of bacteria from xylem sap**. To isolate strains, four gradient dilutions (10⁻³, 10⁻⁴, 10⁻⁵, and 10⁻⁶) of xylem sap were incubated on TSB, R2A, and Ashby's Nitrogen-Free Agar media for 5–7 days at 30 °C (Supplementary Data 9). After incubation, colonies were selected based on their character and colony morphology and were purified by triple serial colony isolation. The isolates were subjected to Sanger sequencing and identified on the basis of PCR analyses with 27 F and 1492 R primers, and alignment against reference. 16S rRNA gene sequences using the BLAST algorithm. Isolates belonging to the core taxa were identified by comparing the 16 S rRNA V5–V7 regions against the highly abundant OTUs (> 0.01%) using UCLUST with 98.65% similarity;[58] this threshold has been reported to accurately distinguish two species. Cladograms were visualised by iTOL.v6.4[59]. The isolated cultures were stored in 30% (v/v) glycerol.

**Nitrogen-fixing capacity of bacterial isolates**. We used three different methods to evaluate the N-fixing capacity of bacterial isolates. First, we observed growth on Ashby's N-Free medium, and documented which strains grew well after streaking of diluted cultures. Then, these strains were analysed by PCR to detect *nifH* with the PolF/PolR primer set[56]. The positive control was the N-fixing strain, *Azotobacter chroococcum* ACCC10006 (Agricultural Culture Collection of China). Strains that did not yield a PCR product with this primer set were analysed using other nitrogenase gene primers including *nifH*-F/*nifH*-R[60] primers and the nested PCR primers FGPH19/PolR (outer primers) and PolF/AQER (inner primers)[56] (Supplementary Table 5). We also conducted acetylene reduction assays (ARA)[61–63] to quantify the nitrogenase activity of putative N-fixing strains. Each tested strain was initially incubated overnight in TSB medium and then washed twice with sterile 0.9% NaCl solution. After centrifugation and re-suspension, the

bacterial pellet was added to a 20 mL serum vial containing 5 mL Dobereiner's N-free liquid medium (Supplementary Data 9), reaching a final $OD_{600}$ of ~0.1. The vials were first flushed with argon to evacuate air, and then 1% and 10% of the headspace was replaced with pure and fresh $O_2$ and $C_2H_2$, respectively. After incubation at 30 °C for 12 h, the gas phase was analysed with a gas chromatograph (Agilent Technologies 6890 N). Data are presented as mean values from five replicate cultures. To test the hypothesis that the core non-N-fixers might assist N-fixation by modifying the oxygen concentration, the nitrogenase activity was measured as described above except that the headspace atmosphere in the sealed vial was not adjusted to 1% $O_2$ by flushing with argon gas so that the initial oxygen concentration was that found in ambient air.

**Draft whole-genome sequencing of cross-referenced core strains.** Isolated genomic DNA was extracted with a TIANamp Bacteria DNA Kit (Tiangen Biotech, Beijing, China). The purified genomic DNA was used to construct a sequencing library, which was generated using the NEB Next® Ultra™ DNA Library Prep Kit for Illumina (NEB, Beverly, MA, USA) following the manufacturer's recommendations. Pooled libraries were sequenced on the NovaSeq-PE 150 platform. After trimming low-quality reads by fastq[64], the clean reads were assembled into draft genomes (excluding contigs of < 300 bp) by SPAdes 3.13.1[65]. Gene prediction and annotation were performed by NCBI PGAP[66], and the putative genes were further annotated by searching against the eggNOG database[67] by emaper. The functional mapping and analysis pipeline (FMAP 0.15)[68] was used to align the filtered reads using BLAST against a KEGG Filtered UniProt[69] reference cluster (e < 1e-5, identity > 70%) and to calculate the number of reads mapping to each KEGG Orthologous group (KO). Other data manipulation was performed using perl scripts developed in-house.

**Potted plant experiment.** Two potted plant experiments were performed to (1) reproduce the endophytic behaviour of isolates from xylem sap; and (2) verify their N-fixation potential in maize. We constructed SynComs consisting of two diazotrophs (K. variicola MNAZ1050 and Citrobacter **sp**. MNAZ1397) and two non-N-fixers (Acinetobacter **sp**. ACZLY512 and R. epipactidis YCCK550) based on a N-fixing capacity test. Each individual strain was cultured overnight in TSB medium at 30 °C and 180 rpm, then cells were collected by centrifugation and the pellet was suspended in sterile 0.9% NaCl solution. Four bacterial suspensions were mixed in equal amounts to a final $OD_{600}$ of ~0.2.

The potted plants were grown in plastic pots filled with loose a soilless mixture consisting of perlite and vermiculite (sterilised by autoclaving). The nutrients needed for plant growth were added as base fertilisers (Supplementary Data 10). The seeds of maize "Zhengdan 958" were surface-disinfected for 15 min with sodium hypochlorite (approximately 2% active chlorine, with 200 µL Tween 80) and washed for 5 min, five times, with sterile water. The final rinse water (100 µL) was spread on TSA medium to check for other attached bacteria. Seeds were allowed to germinate, and then germinated seeds with similar primary root lengths were selected for inoculation with SynComs. Maize plants were grown in a greenhouse under a 16-h light/8-h photoperiod at 30 °C/25 °C (day/night).

**Colonisation of maize tissues by GFP-tagged SynComs.** To verify the endophytic behaviour, each of the four members of SynComs was tagged with green fluorescent protein (GFP) (vector pCPP6529-GFPuv). One GFP-tagged and three other wild bacterial suspensions were mixed as described above, giving a total of four different GFP-tagged combinations. The control was SynComs with no GFP tags. For inoculation, maize seedlings with the endosperm removed were soaked in four GFP-tagged combined solutions for 30 min. The same bacterial suspension was applied to the potted plants at days 10 and 30 after transplanting. After 63 days, stem samples from between the 2nd and 3rd nodes were surface-sterilised with 70% ethanol, collected, and then sectioned using a Leica VT 1000 S vibratome (Leica, Nussloch, Germany). Thin sections (60 µm) and xylem sap were observed under a confocal laser scanning microscope (CLSM, Zeiss LSM 880 confocal microscope, Jena, Germany).

**¹⁵N isotope dilution method.** To verify the N-fixation potential of endophytes in maize, the N fertiliser was replaced with ¹⁵N-labelled $(NH_4)_2SO_4$ (30 % ¹⁵N atom, Shanghai Research Institute of Chemical Industry, China). Maize seedlings with endosperm removed were soaked in a bacterial suspension of SynComs (Treatment group) or autoclaved SynComs (Control group) for 30 min. The same SynComs, active or autoclaved, were re-applied to the potted plants at 10 and 30 days after transplanting, as described above. The roots, stems and leaves were harvested separately from each treatment (four replicates) on day 65. The roots were washed with deionised water to remove adhering isotope residues. The N content and ¹⁵N enrichment of plant tissue were determined using Elementar vario PYRO cube elemental analyser (Vario PYRO Cube, Elementar, Hanau, Germany) and Isoprime 100 isotope mass spectrometer (Isoprime, Cheadle, United Kingdom). The plants inoculated with autoclaved SynComs were used as the reference to calculate BNF with the following equations[34]:

$$\%Ndfa = (1 - \%^{15}Na.e._I / \%^{15}Na.e._{UI}) \times 100 \qquad (2)$$

$$N_2\text{-fixed} = \left(1 - \%^{15}Na.e._I / \%^{15}Na.e._{UI}\right) \times N\ yield._I \qquad (3)$$

where %Ndfa is the percentage of N derived from air, %¹⁵Na.e. (%¹⁵N atom excess) is the enrichment in plants inoculated with SynComs (I) and autoclaved SynComs (UI), $N_2$-fixed is N derived from air, and N yield.I is the total N content of the whole inoculated plant.

**Reporting summary.** Further information on research design is available in the Nature Research Reporting Summary linked to this article.

## Data availability

Raw 16 S rRNA gene amplicon sequence data have been deposited in the Sequence Read Archive under BioProject PRJNA795590. The Whole Genome Shotgun data has been deposited at DDBJ/ENA/GenBank under the accession PRJNA798255. The SILVA database is available at https://www.arb-silva.de. The eggNOG database is available at http://eggnog5.embl.de/download/eggnog_5.0/. KEGG Filtered UniProt reference cluster is available at https://ftp.uniprot.org/pub/databases/uniprot/uniref/uniref90/. Source data are provided with this paper.

## Code availability

All scripts are available in GitHub (https://github.com/PlantNutrition/Liyu).

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

## Acknowledgements

We thank Peng Xu for helpful advice regarding assembly, annotation and interpretation of the genome of core strains, and Yongliang Yan, PhD, and Xiaoxia Zhang, PhD, for assistance in laboratory work. We also thank Jennifer Smith, PhD, from Liwen Bianji (Edanz) (www.liwenbianji.cn/), for editing the English text of a draft of this manuscript. This work was financially supported by the Young Elite Scientists Sponsorship Program by CAST (2017QNRC001).

## Author contributions

C.A. and L.P. conceived the study and supervised the project. L.Z., C.A., M.Z., S.H. and X.X. performed the experiments. L.L., Q.G., Y.W., S.Z., S.H., L.Y., Y.W., K.L., X.Y., D.L. and L.Z. coordinated field experiments. H.W. provided the GFP-containing plasmid. W.Z. and P.H reviewed the manuscript. L.Z, C.A. and L.P. wrote the manuscript.

## Competing interests

The authors declare no competing interests.
