## [Peer Review File · Nature Communications]

Reviewers' Comments:

Reviewer #1:

Remarks to the Author:

I reviewed the research article "A highly conserved bacterial microbiota with nitrogen fixation capacity inhabits the xylem sap in maize plants" by Zhang et al. submitted to Nature Communication. The research performed a detailed census of bacterial communities residing in different maize plant parts across soil types, climate zones, and genotypes. While the bacterial community of root, rhizosphere, and phyllosphere is well studied, we have little information on the microbiome of the vascular tissues. The results showed a core xylem microbiome that is highly conserved and originates from stem, leaf, and root endophytes. Gene functional analysis using FAPROTAX demonstrated the enrichment of N-fixing bacterial species in vascular tissues compared to the belowground compartments. This result was supported by qPCR analysis. Core bacteria from the xylem were isolated and tested for the ability to fix nitrogen. A synthetic microbial community consisting of the most effective N fixers with two helper bacteria was able to colonize the vascular tissue and contribute to nitrogen use efficiency.

I thoroughly enjoyed reading the paper. The experiments are thoughtful, robust, and elegant. This study fulfills the entire loop from causation to the causal impact of the microbiome on plant interactions. A range of tools, techniques, and analyses were performed, which is highly commendable. The papers read well. The research advances microbiome science and fills the critical knowledge gap on the role of vascular microbiome in plant performance. I have few suggestion/comments that will further strengthen the MS.

Introduction:

1. The concept of the 'plant holobiont' is highly controversial. In my opinion, there is very little (if any) evidence suggesting co-evolution between the plant host and comensal microbiota members. There is, on the other hand, abundant studies that show an extremely low (almost undetectable) heritability of the root and leaf microbiota across multiple host species (Bulgarelli et al., 2015 for barley, Wagner et al., 2017 for the Arabidopsis leaf microbiota, Schlaeppli et al., 2014 for the root microbiota; Pfeiffer et al, 2014 for maize, etc.). Although plants co-evolve with some pathogens and symbionts, there is currently no published data (to my knowledge) indicating that they might also co-evolve with their microbiotas as a whole. Because of this, in my opinion, I suggest toning down the 'holobiont' concept.
2. The most exciting part of this paper is the exploration of vascular microbiome and development of SynComs with helper bacteria. I would like to see more information on both aspects.

Results:

1. The use of vascular and xylem creates confusion. Both are interchangeably used throughout the MS. As phloem also belongs to vascular tissue not studied (may be with endophytes), I suggest using Xylem in the results, discussion, and figures.
2. Line 87: The subsequent paragraph provides a general description of microbiota present in different compartments and the impact of soil types, fertilization, and genotypes. I suggest changing the heading and use when you are more focused on the core xylem microbiome.
3. FAPROTAX is a powerful tool to study the functional potential of the microbiome. But it has several limitations. Although authors have validated the findings of FAPROTAX with the qPCR analysis, I still suggest stating the limitations of FAPROTAX upfront briefly. Then say that FAPROTAX is suitable for generating a hypothesis, but additional research must be performed (as done in this paper). Also, see my comments for Figure 3b below.
4. Line 206-215: Were the population counts confirmed by sequence-based methods? If yes, consider including the data.
5. Line 223: Was yield also quantified?
6. Was the increased NUE due to the xylem colonization only, or was it due to the increased abundance in all the plant tissues? The selected bacterial strains are known to possess a range of plant growth promotion ability and other mechanisms in addition to N fixation might be in play leading to increased plant performance.

Discussion:

1. Line 240: The use of word infertile is odd. Just say control
2. Lines 240-243: Can you also check the abundance of N-fixing bacterial strains in across different fertilization regimes? You can use the 2 N fixers in the SynCom as proxy. This will further

strengthen the argument.

3. One of the major advances of the research is the inclusion of helper bacteria in the SynCom. Most of the times SynComs are selected for a particular trait. This point of using the helper bacteria should be discussed in some details.

Figures:

1. Figure 1: Change the caption in bold. The figure shows the general microbiome trends and not the conserved vascular microbiota.
2. Figure 2b. Nitrogen cycling capacity. Is this the FAPROTAX data? Can you clarify the FAPROTAX predict the number of OTU for a trait or number of genes for the trait? Nitrogen cycling capacity can contain many genes for nitrogen cycling. Does OTUs mean that all the genes are present?

Reviewer #2:

Remarks to the Author:

The manuscript describes the core bacterial microbiota with nitrogen-fixation capacity is conserved in the xylem sap of maize plants. Maize plants were grown in six experimental sites with three soil types (black soils, red soils, and fluvo-aquic soils) in China and the soils in these areas have been fertilised for at least 29 years according to three regimes (no fertilisation, Control; chemical fertiliser N, phosphorus, and potassium, NPK; and organic manure plus chemical fertiliser, NPKM). Analyses of bacterial communities from the bulk soil (BS), rhizosphere soil (RS), root endosphere (RE), xylem sap (VE), stem endosphere (SE), leaf endosphere (LE), and phylloplane (P) by 16S rRNA amplicon sequencing of the V5–V7 region suggested that plant compartments were the main driver of maize microbiome composition. Gammaproteobacteria were predominant in stem vascular tissue (VE). The proportion of bacterial taxa carrying the nitrogenase gene (*nifH*) was larger in stem vascular tissues than in other organs such as root and leaf endosphere. Of the 25 core bacterial taxa identified in xylem sap, several isolated strains were confirmed to be active nitrogen-fixers or to assist with biological nitrogen fixation (BNF). Two strains *Klebsiella variicola* MNAZ1050 and *Citrobacter* sp. MNAZ1397 harboured the entire *nif* gene cluster in their genomes were isolated and their combination hereafter referred to KC). Two synthetic communities (SynComs) consist of two core diazotrophs and two non-N-fixers were constructed. The two SynComs were the combinations of KC + *Acinetobacter* sp. ACZLY512 and KC + *Rosenbergiella epipactidis* YCCK550. The two non N-fixers have strong respiratory metabolism and thus may create a micro-oxygen environment conducive to biological N-fixation. GFP-tagged strains and ¹⁵N isotopic dilution method demonstrated that these SynComs do thrive and contribute, through BNF, 11.8% of the total N accumulated in maize stems.

The work is very is of great importance in utilizing bacterial microbiota to improve maize growth and yields. I suggest that the paper can be published after revision.

Comments:

1. Please explain how to assay nitrogenase activity in the air condition. Even if there is non N-fixer, *Klebsiella variicola* and *Citrobacter* sp. Shall not have nitrogenase activity in presence of air. Of course, some N-fixers can have nitrogenase in presence of air when biofilm was produced by themselves.
2. In Fig. 4d, a key gene *nifK* in the *nif* (nitrogen fixation) gene cluster of *Klebsiella variicola* MNAZ1050 was missing. If there is no *nifK* gene, the nitrogenase complex can not be assembled and of course there is no nitrogenase activity. Please check the genome sequence to be sure if there is *nifK* or not.

Minor comments:

1. *nifH* should be in full italics, i.e. *nifH*.
2. Line 195. *Acinetobacter* sp. ACZLY512. Line 208 *Citrobacter* sp. MNAZ1397. And in many place in the manuscript. The sp. should be in bold.
3. Reference part, there are a lot of errors. Please correct these errors.
4. The reference should be included: Yongbin Li et al. (2021) Diazotroph *Paenibacillus triticisoli* BJ-18 drives the variation in bacterial, diazotrophic and fungal communities in the rhizosphere and

root/shoot endosphere of maize. *Int. J. Mol. Sci.* 2021, 22, 1460.

REVIEWER COMMENTS

Reviewer #1 (Remarks to the Author):

I reviewed the research article “A highly conserved bacterial microbiota with nitrogen fixation capacity inhabits the xylem sap in maize plants” by Zhang et al. submitted to Nature Communication. The research performed a detailed census of bacterial communities residing in different maize plant parts across soil types, climate zones, and genotypes. While the bacterial community of root, rhizosphere, and phyllosphere is well studied, we have little information on the microbiome of the vascular tissues. The results showed a core xylem microbiome that is highly conserved and originates from stem, leaf, and root endophytes. Gene functional analysis using FAPROTAX demonstrated the enrichment of N-fixing bacterial species in vascular tissues compared to the belowground compartments. This result was supported by qPCR analysis. Core bacteria from the xylem were isolated and tested for the ability to fix nitrogen. A synthetic microbial community consisting of the most effective N fixers with two helper bacteria was able to colonize the vascular tissue and contribute to nitrogen use efficiency.

I thoroughly enjoyed reading the paper. The experiments are thoughtful, robust, and elegant. This study fulfills the entire loop from causation to the causal impact of the microbiome on plant interactions. A range of tools, techniques, and analyses were performed, which is highly commendable. The papers read well. The research advances microbiome science and fills the critical knowledge gap on the role of vascular microbiome in plant performance. I have few suggestion/comments that will further strengthen the MS.

Introduction:

1. The concept of the 'plant holobiont' is highly controversial. In my opinion, there is very little (if any) evidence suggesting co-evolution between the plant host and comensal microbiota members. There is, on the other hand, abundant studies that show an extremely low (almost undetectable) heritability of the root and leaf microbiota across multiple host species (Bulgarelli et al., 2015 for barley, Wagner et al., 2017 for the Arabidopsis leaf microbiota, Schlaeppi et al., 2014 for the root microbiota; Pfeiffer et al, 2014 for maize, etc.). Although plants co-evolve with some pathogens and symbionts, there is currently no published data (to my knowledge) indicating that they might

also co-evolve with their microbiotas as a whole. Because of this, in my opinion, I suggest toning down the 'holobiont' concept.

2. The most exciting part of this paper is the exploration of vascular microbiome and development of SynComs with helper bacteria. I would like to see more information on both aspects.

Results:

1. The use of vascular and xylem creates confusion. Both are interchangeably used throughout the MS. As phloem also belongs to vascular tissue not studied (may be with endophytes), I suggest using Xylem in the results, discussion, and figures.

2. Line 87: The subsequent paragraph provides a general description of microbiota present in different compartments and the impact of soil types, fertilization, and genotypes. I suggest changing the heading and use when you are more focused on the core xylem microbiome.

3. FAPROTAX is a powerful tool to study the functional potential of the microbiome. But it has several limitations. Although authors have validated the findings of FAPROTAX with the qPCR analysis, I still suggest stating the limitations of FAPROTAX upfront briefly. Then say that FAPROTAX is suitable for generating a hypothesis, but additional research must be performed (as done in this paper). Also, see my comments for Figure 3b below.

4. Line 206-215: Were the population counts confirmed by sequence-based methods? If yes, consider including the data.

5. Line 223: Was yield also quantified?

6. Was the increased NUE due to the xylem colonization only, or was it due to the increased abundance in all the plant tissues? The selected bacterial strains are known to possess a range of plant growth promotion ability and other mechanisms in addition to N fixation might be in play leading to increased plant performance.

Discussion:

1. Line 240: The use of word infertile is odd. Just say control

2. Lines 240-243: Can you also check the abundance of N-fixing bacterial strains in across different fertilization regimes? You can use the 2 N fixers in the SynCom as proxy. This will further strengthen the argument.

3. One of the major advances of the research is the inclusion of helper bacteria in the SynCom. Most of the times SynComs are selected for a particular trait. This point of using the helper bacteria should be discussed in some details.

Figures:

1. Figure 1: Change the caption in bold. The figure shows the general microbiome trends and not the conserved vascular microbiota.
2. Figure 2b. Nitrogen cycling capacity. Is this the FAPROTAX data? Can you clarify the FAPROTAX predict the number of OTU for a trait or number of genes for the trait? Nitrogen cycling capacity can contain many genes for nitrogen cycling. Does OTUs mean that all the genes are present?

Reviewer #2 (Remarks to the Author):

The manuscript describes the core bacterial microbiota with nitrogen-fixation capacity is conserved in the xylem sap of maize plants. Maize plants were grown in six experimental sites with three soil types (black soils, red soils, and fluvo-aquic soils) in China and the soils in these areas have been fertilised for at least 29 years according to three regimes (no fertilisation, Control; chemical fertiliser N, phosphorus, and potassium, NPK; and organic manure plus chemical fertiliser, NPKM). Analyses of bacterial communities from the bulk soil (BS), rhizosphere soil (RS), root endosphere (RE), xylem sap (VE), stem endosphere (SE), leaf endosphere (LE), and phylloplane (P) by 16S rRNA amplicon sequencing of the V5–V7 region suggested that plant compartments were the main driver of maize microbiome composition. Gammaproteobacteria were predominant in stem vascular tissue (VE). The proportion of bacterial taxa carrying the nitrogenase gene (*nifH*) was larger in stem vascular tissues than in other organs such as root and leaf endosphere. Of the 25 core bacterial taxa identified in xylem sap, several isolated strains were confirmed to be active nitrogen-fixers or to assist with biological nitrogen fixation (BNF). Two strains *Klebsiella variicola* MNAZ1050 and *Citrobacter* sp. MNAZ1397 harboured the entire *nif* gene cluster in their genomes were isolated and their combination hereafter referred to KC). Two synthetic communities (SynComs) consist of two core diazotrophs and two non-N-fixers were constructed. The two SynComs were the combinations of

KC + *Acinetobacter* sp. ACZLY512 and KC + *Rosenbergiella epipactidis* YCCK550. The two non N-fixers have strong respiratory metabolism and thus may create a micro-oxygen environment conducive to biological N-fixation. GFP-tagged strains and ¹⁵N isotopic dilution method demonstrated that these SynComs do thrive and contribute, through BNF, 11.8% of the total N accumulated in maize stems.

The work is very is of great importance in utilizing bacterial microbiota to improve maize growth and yields. I suggest that the paper can be published after revision.

Comments:

1. Please explain how to assay nitrogenase activity in the air condition. Even if there is non N-fixer, *Klebsiella variicola* and *Citrobacter* sp. Shall not have nitrogenase activity in presence of air. Of course, some N-fixers can have nitrogenase in presence of air when biofilm was produced by themselves.
2. In Fig. 4d, a key gene *nifK* in the *nif* (nitrogen fixation) gene cluster of *Klebsiella variicola* MNAZ1050 was missing. If there is no *nifK* gene, the nitrogenase complex can not be assembled and of course there is no nitrogenase activity. Please check the genome sequence to be sure if there is *nifK* or not.

Minor comments:

1. *nifH* should be in full italics, i.e. *nifH*.
2. Line 195. *Acinetobacter* sp. ACZLY512. Line 208 *Citrobacter* sp. MNAZ1397. And in many place in the manuscript. The sp. should be in bold.
3. Reference part, there are a lot of errors. Please correct these errors.
4. The reference should be included: Yongbin Li et al. (2021) Diazotroph *Paenibacillus triticisoli* BJ-18 drives the variation in bacterial, diazotrophic and fungal communities in the rhizosphere and root/shoot endosphere of maize. *Int. J. Mol. Sci.* 2021, 22, 1460.

Responses to Reviewers' comments

Reviewer #1 (Remarks to the Author):

I reviewed the research article “A highly conserved bacterial microbiota with nitrogen fixation capacity inhabits the xylem sap in maize plants” by Zhang et al. submitted to Nature Communication. The research performed a detailed census of bacterial communities residing in different maize plant parts across soil types, climate zones, and genotypes. While the bacterial community of root, rhizosphere, and phyllosphere is well studied, we have little information on the microbiome of the vascular tissues. The results showed a core xylem microbiome that is highly conserved and originates from stem, leaf, and root endophytes. Gene functional analysis using FAPROTAX demonstrated the enrichment of N-fixing bacterial species in vascular tissues compared to the belowground compartments. This result was supported by qPCR analysis. Core bacteria from the xylem were isolated and tested for the ability to fix nitrogen. A synthetic microbial community consisting of the most effective N fixers with two helper bacteria was able to colonize the vascular tissue and contribute to nitrogen use efficiency.

I thoroughly enjoyed reading the paper. The experiments are thoughtful, robust, and elegant. This study fulfills the entire loop from causation to the causal impact of the microbiome on plant interactions. A range of tools, techniques, and analyses were performed, which is highly commendable. The papers read well. The research advances microbiome science and fills the critical knowledge gap on the role of vascular microbiome in plant performance. I have few suggestion/comments that will further strengthen the MS.

Reply: Thank you for your valuable comments and suggestions, which have helped improve our manuscript. Please see our point-by-point responses to these comments below.

Introduction:

1. The concept of the 'plant holobiont' is highly controversial. In my opinion, there is very little (if any) evidence suggesting co-evolution between the plant host and comensal microbiota members. There is, on the other hand, abundant studies that show an extremely low (almost undetectable) heritability of the root and leaf microbiota across multiple host species (Bulgarelli et al., 2015 for barley, Wagner et al., 2017 for the Arabidopsis leaf microbiota, Schlaeppi et al., 2014 for the root microbiota; Pfeiffer et al, 2014 for maize, etc.). Although plants co-evolve with some pathogens

and symbionts, there is currently no published data (to my knowledge) indicating that they might also co-evolve with their microbiotas as a whole. Because of this, in my opinion, I suggest toning down the 'holobiont' concept.

Reply: We appreciate your comments. We agree that the concept of the 'plant holobiont' has yet to be conclusively demonstrated. As suggested, we have modified and shortened the corresponding text as follows: "It is now widely accepted that plants and microorganisms can form complex co-associations governed by specific assembly rules⁷. As such, the recruitment and the selection of host-adapted microorganisms is of importance for crop health and nutrition^{7, 8}. It has even been suggested that plants and their associated microbiome are collectively forming holobionts and therefore should no longer be considered as standalone entities⁹. Recent studies have reported that a "core microbiota" exists within the plant, i.e., a subset of the plant microbiota that is reproducibly associated with a particular crop species across a wide range of scales^{7, 10}." (Lines 53–59 in the revised manuscript).

2. The most exciting part of this paper is the exploration of vascular microbiome and development of SynComs with helper bacteria. I would like to see more information on both aspects.

Reply: We have added more information to the Introduction section in response to your comment as specified below:

Lines 63–68 in the revised manuscript:

"In recent years, the development and construction of synthetic communities (SynComs) has provided functional and mechanistic insights into microbe-host relationships and how these relationships influence plant fitness⁷. For example, inoculation of maize seedlings with a simplified and representative seven-strain SynCom resulted in a stronger biocontrol effect against the phytopathogenic fungus *Fusarium verticillioides* than inoculation with each strain separately, indicating a clear benefit to the host¹⁴."

Lines 75–84 in the revised manuscript: "The vascular tissue acts as an effective long-distance transport system, which is driven by hydrostatic pressure gradients between the root and the shoot¹⁹. This driving force ensures smooth transport of solutes and signals among plant organs²⁰. Moreover, the sizes of the holes in the perforated plates between xylem elements are large enough to allow passage of bacteria^{21, 22}. A few studies have reported that some systemic bacterial colonisers can spread to above-ground plant compartments through transpiration-driven xylem

flow²². Anguita-Maeso *et al.* characterized the bacterial communities inhabiting the olive xylem sap by culture-dependent and independent approaches and found that *Sphingomonas* was the most representative genera²³. Nevertheless, little is known about the functional relationship between the xylem-inhabiting microbiota and plant growth and development.”

Results:

1. The use of vascular and xylem creates confusion. Both are interchangeably used throughout the MS. As phloem also belongs to vascular tissue not studied (may be with endophytes), I suggest using Xylem in the results, discussion, and figures.

Reply: We have changed all instances of vascular to xylem throughout the paper.

2. Line 87: The subsequent paragraph provides a general description of microbiota present in different compartments and the impact of soil types, fertilization, and genotypes. I suggest changing the heading and use when you are more focused on the core xylem microbiome.

Reply: We have modified the heading to “Effects of soil type and fertilisation on the maize microbiome” in response to your comment.

3. FAPROTAX is a powerful tool to study the functional potential of the microbiome. But it has several limitations. Although authors have validated the findings of FAPROTAX with the qPCR analysis, I still suggest stating the limitations of FAPROTAX upfront briefly. Then say that FAPROTAX is suitable for generating a hypothesis, but additional research must be performed (as done in this paper). Also, see my comments for Figure 3b below.

Reply: Thank you for your suggestion. We have added the following description to the Results section:

Lines 174–177 in the revised manuscript: “However, like other tools assigning ecological functions based on 16S data, FAPROTAX has some limitations related to the number of reference strains and the affiliation of functions that are poorly conserved; therefore, our FAPROTAX prediction needs to be confirmed by other methods.” Also, please see our response to your comment about Figure 3b below.

4. Line 206-215: Were the population counts confirmed by sequence-based methods? If yes, consider including the data.

Reply: The population counts were not determined by sequence-based methods. The objectives of the experiment described in this section were to (i) reproduce the endophytic behaviour of isolates

from xylem sap, and (ii) verify their N-fixation potential in maize. We ended the experiment when the successful colonisation and nitrogen-fixing capacity of SynComs in the xylem of maize plants were verified by the GFP tracer and ¹⁵N isotopic dilution method, respectively.

5. Line 223: Was yield also quantified?

Reply: We ended the experiment on day 65 after harvesting the roots, stems and leaves of the inoculated maize plants separately; therefore, we do not have yield data. However, our dry matter data suggest that SynComs increased the root dry biomass and root/shoot ratio of maize plants at 65 days.

6. Was the increased NUE due to the xylem colonization only, or was it due to the increased abundance in all the plant tissues? The selected bacterial strains are known to possess a range of plant growth promotion ability and other mechanisms in addition to N fixation might me in play leading to increased plant performance.

Reply: We believe you mean ‘BNF’ data because there are no ‘NUE’ data in the original manuscript.

We agree that there is a multi-mechanism synergistic interaction between the SynComs and their host. For example, we found that the genome of *Klebsiella pneumoniae* MNAZ1050 contained the *ipdC* gene involved in the indole-3-pyruvate (IPyA) pathway, which could produce indole-3-acetic acid (IAA). We identified the *acdS* gene in the genome of *Acinetobacter sp.* ACZLY512. This gene encoded ACC deaminase when expressed, which interfered with ethylene biosynthesis and thus promoted plant growth. The corresponding discussion can be found in lines 290–295 of the revised manuscript.

Our triple-qPCR results (Fig. 3c) showed that the ratio of the *nifH* gene to 16S rRNA in the xylem was highest among the seven compartments in the control soil. Additionally, the ¹⁵N-isotope-enrichment experiment (Table 1) demonstrated that the BNF rates in the stem (11.81%) were about three times higher than those in the root (4.01%). The BNF rate in the leaf was also very low (0.46%). Thus, we suggest that xylem is a key niche for BNF; however, we do not rule out the possibility of translocation of BNF products in maize plants.

Discussion:

1. Line 240: The use of word infertile is odd. Just say control

Reply: “infertile” has been replaced with “control” throughout the revised manuscript.

2. Lines 240-243: Can you also check the abundance of N-fixing bacterial strains in across different fertilization regimes? You can use the 2 N fixers in the SynCom as proxy. This will further strengthen the argument.

Reply: Thank you for your suggestion. We have added supplementary Fig. 3 and the following corresponding discussion in Lines 253–255 in the revised manuscript: “This is also confirmed by the fact that the average relative abundances of two core N-fixers in SynCom (*K. variicola* MNAZ1050 and *Citrobacter sp.* MNAZ1397) were significantly higher in the control soil than in two fertilised soils (NPK and NPKM) (Supplementary Fig. 3).”

Supplementary Fig. 3

The average relative abundances of two core N-fixers in SynComs (*K. variicola* MNAZ1050 and *Citrobacter sp.* MNAZ1397) across different fertilisation regimes. Boxplot displays the median and interquartile range. Upper and lower whiskers extend to data no more than 1.5× the interquartile range from upper and lower edge of the box, respectively. Letters indicate significant differences among different fertilisation regimes (Wilcoxon test, $P_{FDR} < 0.05$).

3. One of the major advances of the research is the inclusion of helper bacteria in the SynCom. Most of the times SynComs are selected for a particular trait. This point of using the helper bacteria should be discussed in some details.

Reply: Thank you for your suggestion. We have expanded on this aspect in Lines 296–301 of the discussion as follows: “Such synergistic effects of the bacterial consortium were previously

observed in other plant-bacterial interactions. For example, Santhanam *et al.* showed that a five-strain SynCom was able to effectively protect *Nicotiana attenuata* against fungal sudden-wilt disease via complementary traits, while the individual strains or smaller combinations of them did not³⁶. Overall, our results highlight the limitation of incorporating only the desirable traits into the design of SynComs since this method might overlook helper strains that could improve the overall beneficial activity of the SynComs.”

Figures:

1. Figure 1: Change the caption in bold. The figure shows the general microbiome trends and not the conserved vascular microbiota.

Reply: We have revised the caption as follows: “Effects of soil type and fertilisation on the maize microbiome”.

2. Figure 2b. Nitrogen cycling capacity. Is this the FAPROTAX data? Can you clarify the FAPROTAX predict the number of OTU for a trait or number of genes for the trait? Nitrogen cycling capacity can contain many genes for nitrogen cycling. Does OTUs mean that all the genes are present?

Reply: Yes, the result of Fig. 3b was from the FAPROTAX data. In our study, FAPROTAX generated sub-tables describing nitrogen cycling (e.g., nitrogen fixation, aerobic ammonia oxidation, denitrification, nitrate respiration, nitrite respiration, nitrification and nitrate reduction data). Their sum is the number of OTUs with nitrogen cycling capacity.

To avoid ambiguities, we have revised the figure legend of Fig. 3b to “Number of OTUs capable of nitrogen cycling in soil and plant compartments based on FAPROTAX prediction”.

Reviewer #2 (Remarks to the Author):

The manuscript describes the core bacterial microbiota with nitrogen-fixation capacity is conserved in the xylem sap of maize plants. Maize plants were grown in six experimental sites with three soil types (black soils, red soils, and fluvo-aquic soils) in China and the soils in these areas have been fertilised for at least 29 years according to three regimes (no fertilisation, Control; chemical fertiliser N, phosphorus, and potassium, NPK; and organic manure plus chemical fertiliser, NPKM). Analyses of bacterial communities from the bulk soil (BS), rhizosphere soil

(RS), root endosphere (RE), xylem sap (VE), stem endosphere (SE), leaf endosphere (LE), and phylloplane (P) by 16S rRNA amplicon sequencing of the V5–V7 region suggested that plant compartments were the main driver of maize microbiome composition. Gammaproteobacteria were predominant in stem vascular tissue (VE). The proportion of bacterial taxa carrying the nitrogenase gene (*nifH*) was larger in stem vascular tissues than in other organs such as root and leaf endosphere. Of the 25 core bacterial taxa identified in xylem sap, several isolated strains were confirmed to be active nitrogen-fixers or to assist with biological nitrogen fixation (BNF). Two strains *Klebsiella variicola* MNAZ1050 and *Citrobacter* sp. MNAZ1397 harboured the entire *nif* gene cluster in their genomes were isolated and their combination hereafter referred to KC). Two synthetic communities (SynComs) consist of two core diazotrophs and two non-N-fixers were constructed. The two SynComs were the combinations of KC + *Acinetobacter* sp. ACZLY512 and KC + *Rosenbergiella epipactidis* YCCK550. The two non N-fixers have strong respiratory metabolism and thus may create a micro-oxygen environment conducive to biological N-fixation. GFP-tagged strains and ¹⁵N isotopic dilution method demonstrated that these SynComs do thrive and contribute, through BNF, 11.8% of the total N accumulated in maize stems. The work is very is of great importance in utilizing bacterial microbiota to improve maize growth and yields. I suggest that the paper can be published after revision.

Reply: Thank you for your valuable comments and suggestions, which have improved our manuscript. Please see our point-by-point responses to the comments below.

1. Please explain how to assay nitrogenase activity in the air condition. Even if there is non N-fixer, *Klebsiella variicola* and *Citrobacter* sp. Shall not have nitrogenase activity in presence of air. Of course, some N-fixers can have nitrogenase in presence of air when biofilm was produced by themselves.

Reply: Thank you for your comment. Yes, bacteria with the ability to perform BNF are known to be sensitive to atmospheric oxygen levels. The nitrogenase activity shown in Fig. 4e was measured using classical acetylene reduction assays (ARA) in a 1% O₂ micro-oxygen environment and not in the presence of air. Specifically, N-fixing strains were added to a 20 mL serum vial containing 5 mL Dobereiner's N-free liquid medium, reaching a final OD₆₀₀ of ~0.1. The vials were then flushed with argon to evacuate the air, after which 1% and 10% of the headspace was replaced with pure and fresh O₂ and C₂H₂, respectively.

We found that the genomes of *Acinetobacter sp.* ACZLY512 showed a higher abundance of gene clusters encoding major enzymes involved in the tricarboxylic acid (TCA) cycle. Based on this finding, we speculated that this non-N-fixer had a strong ability to consume oxygen during respiration, creating a micro-oxygen environment conducive to biological N-fixation. Thus, we further tested the nitrogenase activity under the initial air oxygen content. To accomplish this, all procedures were the same as the classical acetylene reduction assays (ARA), except that the headspace atmosphere in the sealed vial was not adjusted to 1% O₂ by flushing with argon gas so that the initial oxygen concentration was that found in ambient air. As shown in Extended Data Fig. 6b, the results supported our hypothesis that the core non-N-fixers might assist N-fixation by regulating oxygen concentrations.

To avoid ambiguities, we changed “air environment” to “initial atmospheric oxygen levels” in the revised manuscript and Extended Data Fig. 6. We also modified the corresponding text in the Methods section (Lines 467–479 in the revised manuscript) to make it clearer as follows: “We also conducted acetylene reduction assays (ARA)^{61, 62, 63} to quantify the nitrogenase activity of putative N-fixing strains. Each tested strain was initially incubated overnight in TSB medium and then washed twice with sterile 0.9% NaCl solution. After centrifugation and re-suspension, the bacterial pellet was added to a 20-mL serum vial containing 5 mL Dobereiner’s N-free liquid medium (Supplementary Table 6c), reaching a final OD₆₀₀ of ~0.1. The vials were first flushed with argon to evacuate air, and then 1% and 10% of the headspace was replaced with pure and fresh O₂ and C₂H₂, respectively. After incubation at 30 °C for 12 h, the gas phase was analysed with a gas chromatograph (Agilent Technologies 6890 N). Data are presented as mean values from five replicate cultures. To test the hypothesis that the core non-N-fixers might assist N-fixation by modifying the oxygen concentration, the nitrogenase activity was measured as described above except that the headspace atmosphere in the sealed vial was not adjusted to 1% O₂ by flushing with argon gas so that the initial oxygen concentration was that found in ambient air.”

2. In Fig. 4d, a key gene *nifK* in the *nif* (nitrogen fixation) gene cluster of *Klebsiella variicola* MNAZ1050 was missing. If there is no *nifK* gene, the nitrogenase complex can not be assembled and of course there is no nitrogenase activity. Please check the genome sequence to be sure if there is *nifK* or not.

Reply: We apologize for missing *nifK* in the *nif* gene cluster of *Klebsiella variicola* MNAZ1050 when combining the images. Figure 4d has been corrected in the revised version. Please see the genome information about the *nif* gene cluster of *Klebsiella variicola* MNAZ1050, in which the *nifK* gene is present.

The nif gene cluster of Klebsiella variicola MNAZ1050							
Locus	Gene	Type	Start	Stop	Length	GC	Function
gene_001626	nifA	CDS	608228	609803	1575	64.8254	nif-specific transcriptional activator NifA
gene_001627	nifB	CDS	609967	611374	1407	66.0270	nitrogenase cofactor biosynthesis protein NifB
gene_001611	nifD	CDS	593186	594635	1449	58.8682	nitrogenase molybdenum-iron protein alpha chain
gene_001615	nifE	CDS	597739	599113	1374	66.6667	nitrogenase iron-molybdenum cofactor biosynthesis protein NifE
gene_001610	nifH	CDS	592288	593170	882	57.8231	nitrogenase iron protein
gene_001609	nifJ	CDS	588369	591885	3516	63.4812	pyruvate:ferredoxin (flavodoxin) oxidoreductase
gene_001612	nifK	CDS	594690	596253	1563	59.1171	nitrogenase molybdenum-iron protein subunit beta
gene_001623	nifM	CDS	605074	605875	801	69.2884	nitrogen fixation protein NifM
gene_001616	nifN	CDS	599122	600508	1386	66.3059	nitrogenase iron-molybdenum cofactor biosynthesis protein NifN
gene_001619	nifS	CDS	602014	603217	1203	64.6717	cysteine desulfurase NifS
gene_001613	nifT	CDS	596292	596511	219	65.2968	putative nitrogen fixation protein NifT
gene_001618	nifU	CDS	601159	601999	840	62.5000	Fe-S cluster assembly protein NifU
gene_001620	nifV	CDS	603232	604375	1143	68.6789	homocitrate synthase

Minor comments:

1. *nifH* should be in full italics, i.e. *nifH*.

Reply: "*nifH*" has been replaced with "*nifH*" throughout the manuscript.

2. Line 195. *Acinetobacter* sp. ACZLY512. Line 208 *Citrobacter* sp. MNAZ1397. And in many place in the manuscript. The sp. should be in bold.

Reply: "*sp.*" has been bolded throughout the paper.

3. Reference part, there are a lot of errors. Please correct these errors.

Reply: We apologize for these errors, which have been corrected in the revised manuscript!

It should be noted here that the data describing global maize production in 2020 were obtained from two websites, which have been cited as follows in the revised manuscript:

“1. World Data Atlas. Available online:

<https://knoema.com/atlas/topics/Agriculture/Crops-Production-Quantity-tonnes/Maize-production>

(accessed on 12 January 2022).

2. FAOSTAT-Crops Data. Available online: <http://www.fao.org/faostat/en/#data/QC> (accessed on 11 January 2022).”

4. The reference should be included: Yongbin Li et al. (2021) Diazotroph *Paenibacillus triticisoli* BJ-18 drives the variation in bacterial, diazotrophic and fungal communities in the rhizosphere and root/shoot endosphere of maize. *Int. J. Mol. Sci.* 2021, 22, 1460.

Reply: Thank you for your suggestion. This reference has been added in Lines 246–249 of the discussion section in the revised manuscript: “It can therefore be assumed that these facultative endophytes positively selected by the host have the physiological plasticity to thrive in different plant compartments, with functionally important roles^{29,30}.”

Reviewers' Comments:

Reviewer #1:

Remarks to the Author:

I want to congratulate the authors for the incredible research on the microbiota of xylem sap. The authors have addressed all the suggested comments. The addition of new data (Supplementary Fig 3) has further strengthened the argument on the effective colonization of the SynComs members. The MS is significantly improved, and the results will provide a way forward to harness plant-microbiome interactions for increased plant performance.

Reviewer #2:

Remarks to the Author:

The manuscript with title "A highly conserved core bacterial microbiota with nitrogen-fixation capacity inhabits the xylem sap in maize plants" has been greatly improved, The method of nitrogenase activity assay has been described in detail. The nifK gene is added in Fig. 4d. I suggest that the manuscript should be accepted and published by this journal with a minor revision.

Minor comments

Please revise the "sp.". sp. should not be italicized. For example, in *Acinetobacter* sp., *Acinetobacter* should be italicized, but sp. should be in bold.

Reviewer comments, second round

Reviewer #1 (Remarks to the Author):

I want to congratulate the authors for the incredible research on the microbiota of xylem sap. The authors have addressed all the suggested comments. The addition of new data (Supplementary Fig 3) has further strengthened the argument on the effective colonization of the SynComs members. The MS is significantly improved, and the results will provide a way forward to harness plant-microbiome interactions for increased plant performance.

Reviewer #2 (Remarks to the Author):

The manuscript with title "A highly conserved core bacterial microbiota with nitrogen-fixation capacity inhabits the xylem sap in maize plants" has been greatly improved, The method of nitrogenase activity assay has been described in detail. The *nifK* gene is added in Fig. 4d.

I suggest that the manuscript should be accepted and published by this journal with a minor revision.

Minor comments:

Please revise the "sp.". *sp.* should not be italicized. For example, in *Acinetobacter sp.*, *Acinetobacter* should be italicized, but *sp.* should be in bold.

Responses to Reviewers' comments

Reviewer #1 (Remarks to the Author):

I want to congratulate the authors for the incredible research on the microbiota of xylem sap. The authors have addressed all the suggested comments. The addition of new data (Supplementary Fig 3) has further strengthened the argument on the effective colonization of the SynComs members. The MS is significantly improved, and the results will provide a way forward to harness plant-microbiome interactions for increased plant performance.

Reply: Thank you very much!

Reviewer #2 (Remarks to the Author):

The manuscript with title "A highly conserved core bacterial microbiota with nitrogen-fixation capacity inhabits the xylem sap in maize plants" has been greatly improved, The method of nitrogenase activity assay has been described in detail. The nifK gene is added in Fig. 4d.

I suggest that the manuscript should be accepted and published by this journal with a minor revision.

Minor comments:

Please revise the "sp.". sp. should not be italicized. For example, in *Acinetobacter* sp., *Acinetobacter* should be italicized, but sp. should be in bold.

Reply: "sp." has been replaced with "**sp.**" throughout the manuscript. Thank you very much!